



# A Bayesian approach to understanding the key factors influencing temporal variability in stream water quality: a case study in the Great Barrier Reef catchments

Shuci Liu[1*], Dongryeol Ryu[1], J. Angus Webb[1], Anna Lintern[1,2], Danlu Guo[1], David Waters[3], Andrew W. Western[1]

[1]Department of Infrastructure Engineering, The University of Melbourne, Parkville, VIC, 3010.
[2]Department of Civil Engineering, Monash University, VIC, 3800.
[3]Queensland Department of Natural Resources, Mines and Energy, Toowoomba, QLD, 4350.

*Correspondence to*: Shuci Liu (shucil@student.unimelb.edu.au)

**Abstract.** Stream water quality is highly variable both across space and time. Water quality monitoring programs have collected a large amount of data that provide a good basis to investigate the key drivers of spatial and temporal variability. Event-based water quality monitoring data in the Great Barrier Reef catchments in northern Australia provides an opportunity to further our understanding of water quality dynamics in sub-tropical and tropical regions. This study investigated nine water quality constituents, including sediments, nutrients and salinity, with the aim of: 1) identifying the influential environmental drivers of temporal variation in flow event concentrations; and 2) developing a modelling framework to predict the temporal variation in water quality at multiple sites simultaneously. This study used a hierarchical Bayesian model averaging framework to explore the relationship between event concentration and catchment-scale environmental variables (e.g., runoff, rainfall and groundcover conditions). Key factors affecting the temporal changes in water quality varied among constituent concentrations, as well as between catchments. Catchment rainfall and runoff affected in-stream particulate constituents, while catchment wetness and vegetation cover had more impact on dissolved nutrient concentration and salinity. In addition, in large dry catchments, antecedent catchment soil moisture and vegetation had a large influence on dissolved nutrients, which highlights the important effect of catchment hydrological connectivity on pollutant mobilisation and delivery.

## 1 Introduction

In-stream water quality plays a vital role in influencing the health of freshwater ecosystems (Bhaduri et al., 2016; Pérez-Gutiérrez et al., 2017), which in turn underpins environmental, social and economic sustainability (McGrane, 2016; Ustaoğlu et al., 2020). Pollution derived from agricultural land and urban development has led to water quality degradation in streams and lakes in many regions of the world (Novotny, 1999; Peters and Meybeck, 2000; Ren et al., 2003; Sharpley, 2016). Among these water quality issues, coastal regions with high agricultural production have been delivering large amounts of pollutants to the ocean, where marine ecosystems are vulnerable to the evaluated levels of nutrients and sediments





(Carpenter et al., 1998; Gorman et al., 2009). It is estimated that 60% of coastal rivers in the USA have been moderately to severely degraded (Gorman et al., 2009; Howarth et al., 2002). Therefore, to protect both freshwater and marine ecosystems, better management of catchment-derived pollutants is needed.

Surface water quality is highly variable across spatial and temporal scales (Allan et al., 1997; Guo et al., 2019; Lintern et al.,
2018a). These spatial and temporal variations are the result of complex interactions between three key pollutant processes in catchments, namely, sources (e.g., atmospheric deposition or anthropogenic inputs), mobilisation (e.g., detachment from the sources), and delivery (e.g., transport from sources to receiving waters) (Granger et al., 2010; Lintern et al., 2018a). Across different catchments, spatial differences in water quality concentration can vary markedly due, in part, to heterogeneity of natural landscapes in catchments (e.g., geology, topography and climate) and human-induced activities (e.g., agricultural and
urban development) (Liu et al., 2018; Mainali and Chang, 2018; Mainali et al., 2019). At a site, water quality concentrations can also exhibit significant daily, event, seasonal and annual variability, driven by variations in climatic conditions, in-stream biogeochemical processes and hydrological transport (Hill, 1996; Pretty et al., 2006; Thompson et al., 2011). Thus, it can be challenging to design effective catchment water quality management strategies without a sound understanding of the spatial and temporal variation in water quality and the associated driving factors.

While it has been acknowledged that both spatial and temporal variations in water quality are of great importance for effective water resources management (Guo et al., 2020), this study focused on identifying key drivers of the temporal variability in water quality. It follows our previous study investigating spatial variation in water quality in the same region (Liu et al., 2018). A wide range of environmental factors may affect temporal changes in water quality. Runoff and rainfall have been considered as important factors and the most commonly used explanatory variables to describe temporal variation
in water quality (Deletic and Maksimovic, 1998; Kim et al., 2007; Yang et al., 2009), for example early work by Hem (1948), Walling and Foster (1975) and Walling (1984). Studies considering hydrometeorological drivers have been typically related to the mobilisation and delivery of pollutants. Catchment soil moisture and evapotranspiration can also have an important role in determining the hydrological cycle (e.g., runoff generation), such as sediments (Bieger et al., 2014; Varanou et al., 2002), nutrients (Bouraoui et al., 2002; Lam et al., 2010) and salinity (Brevik et al., 2006; Tweed et al.,
2007), thereby affecting the surface water quality. In addition, riverine water quality has been found to be strongly influenced by seasonal changes in vegetation cover (de Mello et al., 2018; Griffith et al., 2002; Shi et al., 2017). For instance, satellite-derived vegetation indices have provided an opportunity to explore the relationship between land cover and water quality temporal dynamics (Fu and Burgher, 2015; Griffith, 2002; Singh et al., 2013; Whistler, 1996). Even though significant research efforts have been made to explore the relationship between water quality and these
environmental conditions, a comprehensive understanding of their relative importance in diverse environments and at large scales is still lacking.

Statistical modelling has been widely used to investigate water quality temporal dynamics in response to changes in the abovementioned environmental factors (Alexander et al., 2002; Fu et al., 2019; Miller et al., 2014; Singh et al., 2013; Zhang and Blomquist, 2018; Zhang et al., 2016; Zhang and Schilling, 2005). However, existing studies have limitations. Firstly,





water quality monitoring data have often been limited to low sampling frequencies, typically using monthly grab samples. This can result in a lack of information on water quality dynamics over runoff/storm events, which is when a significant proportion of nutrients and sediment loads are transported (Lloyd et al., 2016; Sherriff et al., 2015). Secondly, most studies on statistical water quality modelling have only investigated the relationship between water quality and explanatory variables in a single or limited number of catchments in small regions (Chang et al., 2015; Khan et al., 2020; Koci et al., 2020; Liu et

al., 2008b; Noori et al., 2020; Wang et al., 2016; Zhang et al., 2016). Few studies have investigated water quality at multiple locations using the same modelling framework. Lastly, studies have usually relied on a single 'best' model with an assumption that it best approximated the true drivers of water quality (Paliwal et al., 2007; Zhang et al., 2009). This ignores the issue of selection uncertainty. Furthermore, relying on a single model structure might result in misleading conclusions or overconfidence in the results (Link and Barker, 2006; Wintle et al., 2003).

This study attempted to address these knowledge gaps, taking advantages of event-based water quality monitoring data from the Great Barrier Reef (GBR) catchments in northern Australia, where land-derived pollutants have posed threats to ecosystem of the GBR lagoon (Brodie et al., 2012; Hunter and Walton, 2008; McKergow et al., 2005b; Waterhouse et al., 2017). We targeted nine common water quality indicators, including sediments, nutrients and salinity. Bayesian hierarchical modelling was used to investigate water quality spatial and temporal variation. This allowed the prediction of water quality

in multiple catchments, as well as simultaneously quantifying parameter uncertainty (Gelman et al., 2013; Rode et al., 2010; Webb and King, 2009). In addition, we used Bayesian model averaging (BMA) approaches to identify the relative importance of the different environmental factors and provide multi-model weighted predictions, which have been shown to better quantify the uncertainty arising from model selection (Höge et al., 2019; Raftery et al., 1997; Wang et al., 2012). Overall, this study aimed to: (1) identify the key drivers of temporal variation in water quality; and (2) predict water quality

temporal variation using a Bayesian multi-model approach.

## 2 Materials and methods

### 2.1 Study area

The GBR catchments, situated in north-eastern Australia (Figure 1), consist of six natural resource management regions whose streams and rivers discharge into the Great Barrier Reef lagoon. These catchments cover a 437,354 km$^2$,

approximately a quarter of the state of Queensland, and exhibit significant diversity in climatic, geological and topographical landscape characteristics, as well as in land use and land management (Bartley et al., 2018; Gilbert and Brodie, 2001). The GBR catchments range from small, steep, high-energy streams in the wet tropics, which are dominated by sugarcane crops and rainforest, to large inland catchments used for savannah grazing, and crops (e.g., grain) and with extensive low energy floodplains in the dry tropics (Table 1) (Davis et al., 2017; Koci et al., 2019; McKergow et al., 2005a). Spatial and temporal

variations in rainfall in the GBR catchments are a major cause of the diversity in land use patterns. Annual rainfall ranges from less than 500 mm in the south-west to more than 8000 mm in the north-east (Figure 2 [c]) (Davis et al., 2017; Kuhnert





et al., 2009). Distinct wet (November to April) and dry (May to October) seasons result in high seasonal variation in runoff and El Nino-Southern Oscillation (ENSO) leads to high inter-annual variability (Day and McKeon, 2018; Murphy and Ribbe, 2004). In the dry tropics, a few large events in the wet season contribute the majority of annual runoff, and constant low flow dominates during the dry season (Jarihani et al., 2017).

Thirty-two sites within the GBR catchments were selected as case study catchments (Figure 1 and Table C1 in Appendix C). Previous multivariate analysis of the patterns of time-averaged concentrations indicated that there were two groups of sites (Table 1 and Figure 2 [a]), which was a result of spatial heterogeneity in catchment landscape characteristics (Figure 2 [b], [c] and [d]) (Liu et al., 2018).

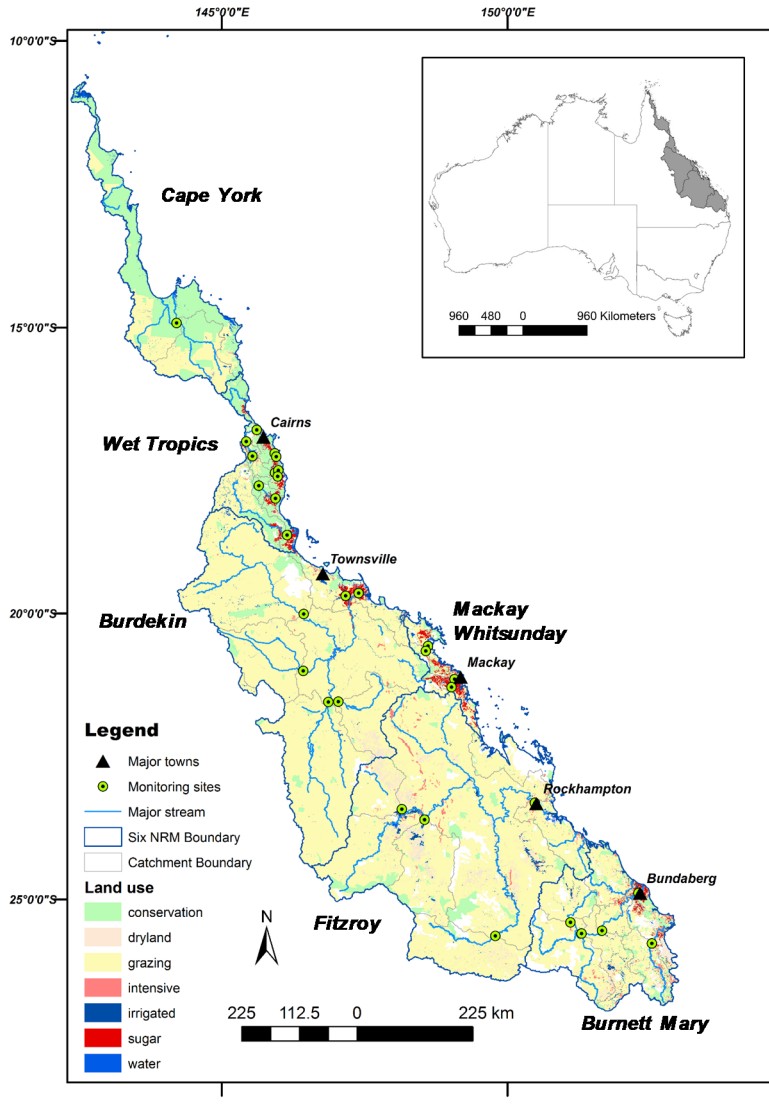

**Figure 1. The Great Barrier Reef catchments, monitoring sites, land uses and the six natural resource management (NRM) regions. Land uses have the following characteristics: (1) conservation (forest, woodland, savannah, etc for conservation**





purposes); (2) dryland (rainfed agriculture including cereals but excluding grazing and sugar cane); (3) grazing (primarily cattle grazing of native and introduced vegetation); (4) intensive (urban areas, roads, etc); (5) irrigated (irrigated cropping excluding
sugar cane); (6) sugar (rain-fed and irrigated sugar cane); and (7) water (water bodies, including lake, river, and marsh/wetland).

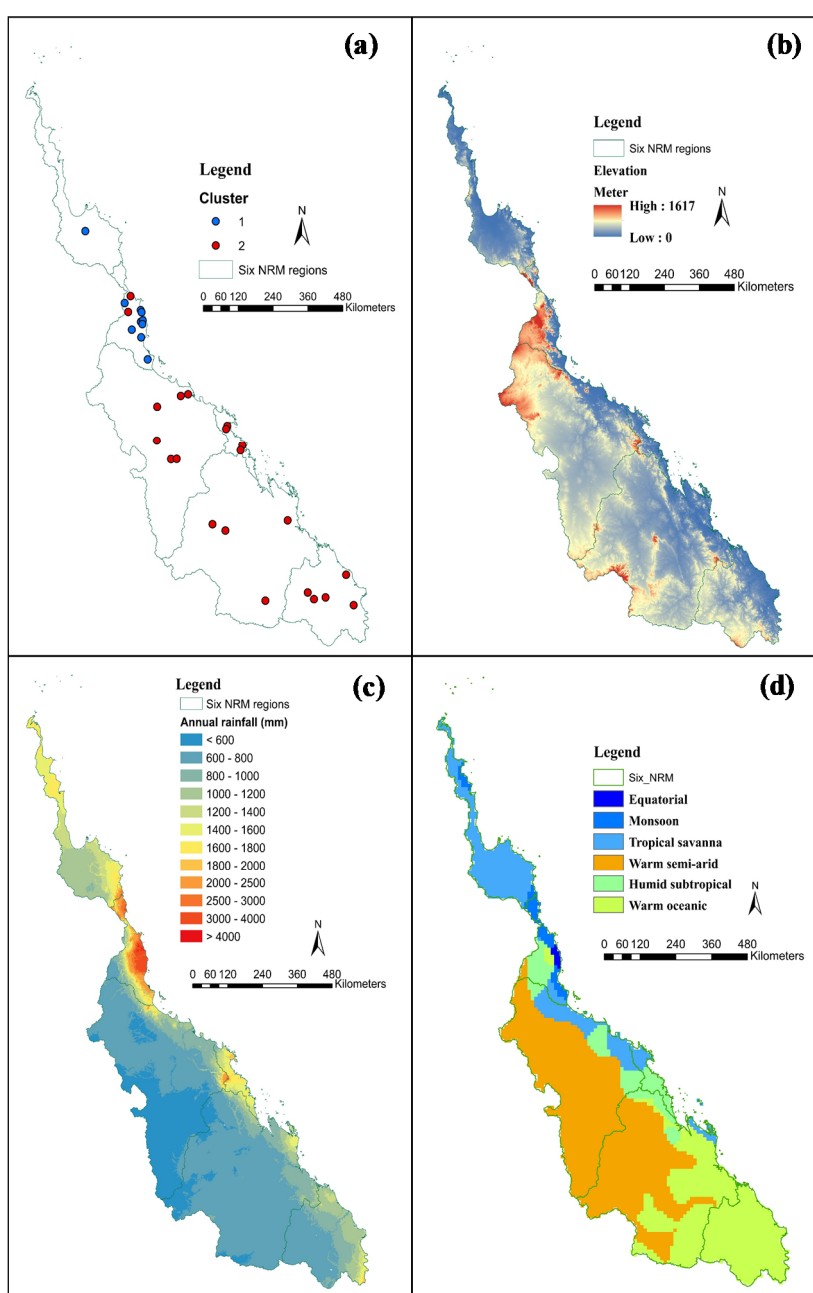

**Figure 2: Spatial information of the GBR catchments in northeast of Australia: [a] site locations showing two groups based on clustering analysis of spatial variability in time-averaged water quality (Liu et al., 2018); [b] topographic elevation (250 m resolution) (Geoscience Australia, 2008); [c] annual average rainfall (Bureau of Meteorology, 2012), and [d] updated Köppen-**
**Geiger climate zone classification (Peel et al., 2007).**





**Table 1: Summary of differences in landscape characteristics between the two clusters of sites (Liu et al., 2018).**

| Cluster | Climate | Hydrology | Land use/land cover | Topography |
|---|---|---|---|---|
| **1** | Wet tropics region with high annual rainfall | Perennial, high energy rivers | Dominated by conservation (e.g. rainforest), and cropping (e.g. sugar) | Small and steep |
| **2** | Mostly dry tropics, relatively dry with clear seasonal variability in rainfall | Ephemeral, low energy rivers, cease-to-flow in dry period | Dominated by brigalow native vegetation, and pastures for grazing, | Large and flat |

## 2.2 Data collection and preparation

### 2.2.1 Water quality data

The nine studied constituents were total suspended solids (TSS), particulate nitrogen (PN), oxidized nitrogen ($NO_X$),
ammonium nitrogen ($NH_4$), dissolved organic nitrogen (DON), filterable reactive phosphorus (FRP), dissolved organic phosphorus (DOP), particulate phosphorus (PP), and electrical conductivity (EC). Water quality monitoring data collected for the 32 GBR catchments over the 11-year period of 2006 to 2016 were obtained from the Loads Monitoring Program (Turner et al., 2012). This dataset contained both high-frequency event-based samples (e.g., daily or every few hours by automatic samplers) that were taken during runoff events, as well as grab samples (e.g., monthly) that were taken under
baseflow conditions (Orr et al., 2014; Waters et al., 2013; Waters and Packett, 2007). As EC data from the Loads Monitoring Program were limited, we extracted additional EC data from the Water Monitoring Information Portal provided by the Department of Natural Resources, Mines and Energy of Queensland (DNRME, 2018) to complement the Loads Monitoring Program records.

### 2.2.2 Event mean concentration

We extracted continuous discharge records for each site from the Water Monitoring Information Portal (DNRME, 2018) to identify individual runoff events. An automated hydrograph analysis tool – *HydRun* (Tang and Carey, 2017) was used to delineate runoff events. The start and end points of a specific event were determined by using a local-minimum method that calculates the first derivative of the streamflow record (separated from baseflow). The event-mean concentration (EMC) was then calculated for each event that had at least two samples on each of the rising and falling limbs of the hydrograph. This
ensured that the water quality dynamics over a runoff event were reasonably well-captured, and that the derived EMCs were reliable (Waters and Packett, 2007). For each event, the EMC of a constituent was calculated as the total load per unit flow volume within the event using (Bartley et al., 2012; Joo et al., 2012):





$$EMC = \frac{EventLoad}{EventFlowVolume} = \frac{\sum_{j=0}^{n} \frac{c_j + c_{j+1}}{2} \times q_{j+1/2} \times t_{j+1/2}}{\sum_{j=0}^{n} q_{j+1/2} \times t_{j+1/2}} \qquad (1)$$

where $n$ is the total number of samples for a given event, $c_j$ is concentration of the $j^{th}$ sample, $q_{j+1/2}$ and $t_{j+1/2}$ are the inter-sample mean discharge and time interval between $j^{th}$ and $(j+1)^{th}$ samples. The concentrations at the start and end of the event ($c_0$ and $c_{n+1}$) are assumed to be the averaged value for samples during baseflow (with baseflow identified in the previous section). The EMCs were essentially flow-weighted mean concentrations over individual runoff events, which allowed the comparison of water quality across catchments with contrasting flow regimes (e.g., two clusters of sites in Figure 2) (Cooke et al., 2000; Richards and Baker, 1993). A total of 1412 events was identified across the 32 sites, and, depending on data availability, EMCs were calculated for between 21% (DOP) and 43% (TSS) of these identified runoff events (Table C2).

The derived EMCs were Box-Cox transformed to improve the symmetry of the response variable (Box and Cox, 1964) to improve model fitting (Hawkins and Weisberg, 2017; Lawrance, 1988; Zhang and Yang, 2017). The site-level Box-Cox transformation parameter λ for each constituent was first identified, using the *car* package in *R* (Fox et al., 2012; R Core Team, 2013). Then, for each constituent, the average λ from the 32 sites was used to transform all available EMCs for that specific constituent. This ensured that an identical transformation parameter was applied across the different sites for each constituent (Guo et al., 2019).

### 2.2.3 Explanatory variables

This study investigated the effect of various hydrologic, climatic and vegetation cover characteristics for different events. These characteristics included runoff, catchment root zone soil moisture, actual evapotranspiration rainfall, air temperature, and vegetation cover. The continuous streamflow monitoring data, gridded weather and climatic products, and remotely sensed imagery were used to derive catchment average conditions for each event (Table 2).

**Table 2. Explanatory variables and their data sources**

| Explanatory variable | Unit | Spatial resolution | Source |
|---|---|---|---|
| Daily runoff | mm/d | point measurements | Queensland Department of Natural Resources, Mines and Energy (DNRME, 2018). Available from https://water-monitoring.information.qld.gov.au/ |
| Daily rainfall | mm | 5 km × 5 km | Australia Water Availability Project (AWAP) (Raupach et al., 2009). Available from http://www.csiro.au/awap/ |
| Daily temperature | °C | | |
| 16-day normalized difference vegetation index (NDVI) | - | 1 km × 1 km | Moderate Resolution Imaging Spectroradiometer (MODIS) - MOD13A2v006 (Didan, 2015). Available from https://earthdata.nasa.gov/ |
| Daily soil moisture (root zone 0 -100 cm) | mm | 5 km × 5 km | Australia Landscape Water Balance model (AWRA-L) (Frost et al., 2016). Available from http://www.bom.gov.au/water/landscape |
| Daily actual ET | Mm | | |



*Note:* ET – evapotranspiration

For individual runoff events identified in the previous section, three groups of event characteristics were prepared,
characterising pre-event, during-event and post-event conditions (Table 3). Except for runoff, data for all explanatory
variables were first extracted from gridded data using catchment boundaries were delineated using the Geofabric tool
provided by the Australian Bureau of Meteorology (Bureau of Meteorology, 2012) (Figure 1). The catchment average time
series data were then averaged over the specific time-window related to the event (Table 3).

**Table 3: Three groups of event characteristics and averaging method**

| Group | Explanatory variable | Abbreviation used in figures and tables in paper | Calculation method |
|---|---|---|---|
| During-event | Average runoff | Event_ave_Q | Average of daily runoff during event |
| | Maximum runoff | Event_max_Q | Maximum of daily runoff during event |
| | Average rainfall | Event_ave_P | Average of daily rainfall during event |
| | Maximum rainfall | Event_max_P | Maximum of daily rainfall during event |
| | Average temperature | Event_T | Average of daily temperature during event |
| | Average NDVI | Event_NDVI | Average of NDVI during event |
| | Average soil moisture | Event_SM | Average of daily soil moisture during event |
| | Average actual ET | Event_AET | Average of daily actual ET during event |
| Pre-event | Average runoff | Ante_Q | Average of daily runoff for 7 days prior to event |
| | Average rainfall | Ante_P | Average of daily rainfall for 7 days prior to event |
| | Average NDVI | Ante_NDVI | Average of NDVI for 3 months prior to event |
| | Average soil moisture | Ante_SM | Average of daily soil moisture for 7 days prior to event |
| | Average actual ET | Ante_AET | Average of actual ET for 7 days prior to event |
| Post-event | Average runoff | Post_Q | Average of daily runoff for 7 days after event |

*Note:* Q – runoff; P – rainfall; T – temperature; NDVI – normalized difference vegetation index; SM – root zone soil moisture; ET –
evapotranspiration.

The explanatory variables in the during-event conditions were averaged over the duration of the event. For the pre-event and
post-event conditions, the 7 days prior to and after the event were used as the time-window (except NDVI). The 7-day period
was the median of the time of concentration (i.e., the time for runoff to travel from the most remote point of the catchment to
the monitoring site) across all catchments. These were estimated from catchment topography using the Bransby-William's
equation, following its wide application in Australian catchments for flood estimation (French et al., 1974; Pilgrim et al.,
1987). The ground cover was quantified by NDVI, an indicator of the biophysical condition of the vegetation canopy
(Griffith et al., 2002). Previous studies have also shown that there is a time-lag between water availability and a change in
ground cover, which is typically three months for Australian catchments (De Keersmaecker et al., 2015; Papagiannopoulou
et al., 2017). Therefore, to represent the pre-event ground cover condition, we averaged all available NDVI measurements



for three months prior to an event. The runoff after the event (7 days) was also included as an indicator of catchment wetness at the end of the event, to assess if hydrologic condition towards the end of an event influences the temporal variation in water quality.

Similar to the EMCs, all the explanatory variables were Box-Cox transformed following the procedure described in Sect. 2.2.2. In addition, prior to the analyses, both transformed EMCs and explanatory variables were standardized to a mean of zero and standard deviation of one. As such, the magnitude of a coefficient indicates the effect of each predictor relative to other predictors (Wan et al., 2014). The cross-correlation (non-parametric Spearman's Rank correlation coefficient) of all transformed predictors is provided in Figure B1, Appendix B.

### 2.3 Modelling: driver identification and water quality prediction using multi-model inference


The statistical analysis and modelling followed several steps (Figure 3). The Bayesian modelling framework was applied to catchments in Clusters 1 and 2 separately. This is because we assumed that the key drivers of temporal variability in water quality would differ between two Clusters due to their differences in land use and climate.

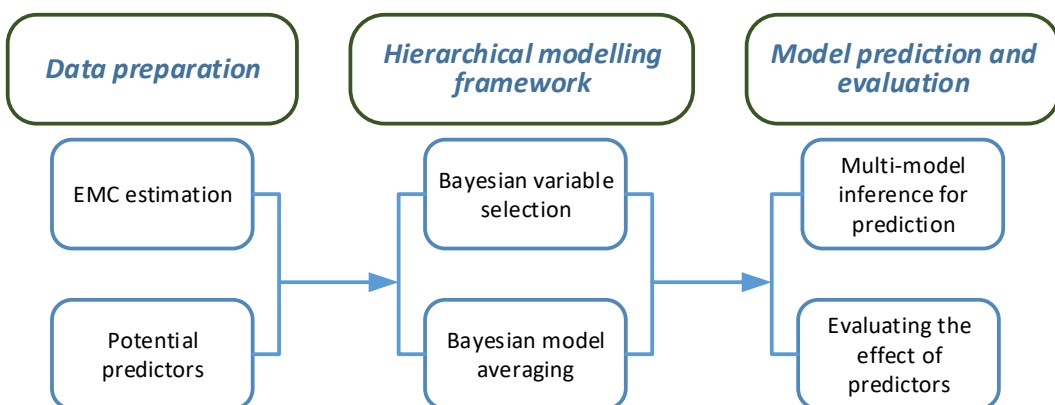


**Figure 3: Analyses steps; the detailed methods used in the hierarchical modelling framework and model prediction and evaluation are in the following sections.**

### 2.3.1 Bayesian variable selection

To investigate the relative importance of individual predictors, an indicator Bayesian variable selection method was used
called Gibbs variable selection (GVS) (George and McCulloch, 1993; Ntzoufras, 2002). An auxiliary inclusion variable $I_n$ (Eq. (2)) for each predictor was introduced to indicate whether that predictor was 'in' or 'out' of an individual iteration of the hierarchical modelling structure.

$$I_n \begin{Bmatrix} 1, & n^{th}\ predictor\ present \\ 0, & n^{th}\ predictor\ absent \end{Bmatrix} \qquad (2)$$



$I_n$ was modelled at the top level of the hierarchy which enabled the use of identical model structures (i.e., combination of predictors) across different sites. The overarching hierarchical modelling framework was defined as follows:

$$y_{i,j} \sim N(\mu_{i,j}, \sigma) \qquad (3)$$

$$\mu_{i,j} = \overline{mean}_j + \overline{std}_j \times \Delta_{i,j} \qquad (4)$$

$$\Delta_{i,j} = \sum_{n=1}^{N} \theta_{n,j} \times x_{n,i,j} \qquad (5)$$

$$\theta_{n,j} = I_n \times \beta_{n,j} \qquad (6)$$

The data-level model (Eq. (3)) assumed that the EMC of a particular constituent (e.g., one of TSS, NOx, EC, etc) at $i^{th}$ time step in the $j^{th}$ sub-catchment, $y_{i,j}$, followed a normal distribution (denoted as $N(\cdot)$), with mean $\mu_{ij}$ and a global standard deviation $\sigma$. The mean value, $\mu_{ij}$ was modelled as the observed site-level averaged EMC $\overline{mean}_j$ plus $\overline{std}_j \times \Delta_{i,j}$, with the latter term being defined as the deviation from this averaged value (Eq. (4)) (Guo et al., 2019). The deviation term incorporated the site-level observed standard deviation $\overline{std}_j$, making $\Delta_{i,j}$ a standardised measure that could be compared

across sites. $\Delta_{i,j}$ was further modelled as a linear additive function (Eq. (5) of all candidate predictors $x_n$ in $n = 1, 2, \ldots, N = 14$ (e.g., event average runoff, rainfall and NDVI). Consequently, $\Delta_{i,j}$ was defined as the temporal variability in water quality, and was the quantity of interest. The effect size ($\theta_{n,j}$) of individual predictors was another latent variable used in the GVS, and was estimated as the product of $I_n$ and the regression coefficient $\beta_{n,j}$ (Eq. (6)), such that $\theta_{n,j}$ was either $\beta_{n,j}$ ($I_n = 1$), or 0 ($I_n = 0$).

**2.3.2 Hierarchical prior specification and Bayesian inference of key drivers**

Bayesian inference required specification of prior distributions for each model parameter. We used a hierarchical conditional prior specification for predictor coefficients, allowing the site-specific parameter values that describe the effects of each of the temporal predictors ($\beta_{1,j}, \beta_{2,j\ldots}, \beta_{n,j}$) to be exchangeable between sites (Liu et al., 2008a; O'Hara and Sillanpää, 2009; Webb and King, 2009). The detail specification of priors for each model parameter can be found in Appendix A. In addition,

to identify key drivers affecting temporal changes in water quality, the posterior inclusion probability (PIP - $P(I_n = 1|y)$, Eq. (A8) in Appendix A) of each predictor was used to compare the relative importance of individual predictors (i.e., how often the $n^{th}$ predictor was 'in' the model).

**2.3.3 Prediction from multi-model inference**

We used Bayesian Model Averaging to generate an ensemble of predictions of temporal variation in EMC for individual

constituents (Eq. (7)). The average posterior distribution of a quantity of interest (i.e., temporal variability in EMC) was





generated using the parameters (e.g., $\beta_{1,j}, \beta_{2,j\ldots}, \beta_{n,j}$) sampled from the posterior distribution to simulate EMC values using the specific model, defined as follows:

$$[\hat{y}|y] = \sum_{x=1}^{L} [\hat{y}|y, M_x]P(M_x|y) \qquad (7)$$

where $[\hat{y}|y, M_x]$ is the posterior distribution of a vector $\hat{y}$ of (prediction) derived from model $M_x$, and $P(M_k|y)$) is the posterior model probability (PMP, Eq. (A8), in Appendix A) (Hooten and Hobbs, 2015; O'Hara and Sillanpää, 2009).

### 2.3.4 Model evaluation and implementation

The proposed modelling framework was applied to the two clusters of sites independently. This allowed an investigation of whether the spatial heterogeneity in catchment landscapes led to differences in the key factors controlling temporal variation in water quality. The key drivers were determined as the predictors with a PIP above 0.8 (i.e., over 80% of the models included these predictors).

To further understand the reliability and robustness of the BMA framework, the consistency of the posterior inclusion probability of individual predictors was investigated by resampling subsets of the observations multiple times (Kohavi, 1995). For each cluster, 80% of events within one site were first randomly selected and the posterior inclusion probability for this subset of observations was estimated. This was repeated 1,000 times to produce a distribution of posterior inclusion probabilities for individual predictors, which was then used to assess the uncertainty in the posterior inclusion probability.

An ensemble of the averaged prediction in temporal variability of each event was obtained from each iteration of parameter updating using Markov chain Monte Carlo (MCMC). The model fit was evaluated using the Nash-Sutcliffe coefficient (NSE) (Nash and Sutcliffe, 1970) between the observed temporal variability and the median of ensemble predictions $\hat{y}$ derived from the BMA (Eq. (7)). The NSE was calculated at both the cluster- and site-levels. The model residuals were also checked for normality and heteroscedasticity (i.e., relationship between the residual and predictors). In addition, model performance was evaluated by providing the 50% and 95% credible interval (CI) of each prediction.

To compare the relative importance of the predictors that have been widely used in existing literature (i.e., runoff and rainfall) and other predictors (e.g., soil moisture, temperature, evapotranspiration, and vegetation cover), the modelling framework was re-calibrated using only the rainfall/runoff related predictors (including all pre-, during- and post-event predictors). This estimated the degree of improvement in the model's explanatory power with the inclusion of environmental variables, such as catchment wetness and ground vegetation cover conditions.

The hierarchical modelling framework was implemented in *JAGS* (Plummer, 2003, 2013a), using the package *rjags* in *R* (Plummer, 2013b; R Core Team, 2013), which enabled both the estimation of parameter values from prior distributions with Markov chain Monte Carlo (MCMC) and the generation of model-averaged predictions. The MCMC sampling had three parallel chains with 25,000 iterations for each chain. The first 5,000 iterations were discarded as a 'burn-in' period to allow





convergence of the Markov chains, resulting in 60,000 values to estimate the posterior distribution for each model parameter and make model predictions.

## 3. Results

### 3.1 Key drivers of temporal variability in water quality

The three key measures that were used to quantify the effect of individual predictors are: (1) estimates of posterior inclusion probability (PIP), which quantifies relative importance of individual predictors; (2) posterior model probability (PMP), which estimates differences in plausible model structures; and (3) posterior distributions of coefficients for the key drivers (i.e., effect size, e.g., $\theta_{1,j}$, $\theta_{2,j...}$, $\theta_{n,j}$ in Eq. (6)), which measures direction and magnitude of the effect of key predictors on water quality temporal variability.

Posterior inclusion probability (Figure 4 and Table C3 in Appendix C) from the Bayesian modelling results indicated that, in general, antecedent vegetation condition and antecedent soil moisture were key factors in explaining temporal variation in water quality, especially for Cluster 2 (warmer, drier) sites. Catchment runoff and rainfall were the second most important group of factors, especially for particulate pollutants (TSS, PN and PP; Clusters 1 and 2) and salinity. In addition, the three groups of predictors (pre-, during-, post-event) showed varying effects among the constituents. With regard to during-event

conditions, event average runoff (*Event_ave_Q*), event maximum runoff (*Event_max_Q*) and event average rainfall (*Event_ave_P*) were three important factors with relatively high PIP. In contrast, among pre-event conditions, antecedent NDVI (*Ante_NDVI*) and antecedent soil moisture (*Ante_SM*) were driving factors for the majority of the constituents. Post-event runoff (*Post_Q*) only affected a few constituents (e.g., on $NO_X$ and FRP for Cluster 2), compared with the other two groups of predictors. Overall, there were notable differences in the important predictors for Clusters 1 and 2, and more

important predictors were found for the Cluster 2 sites.





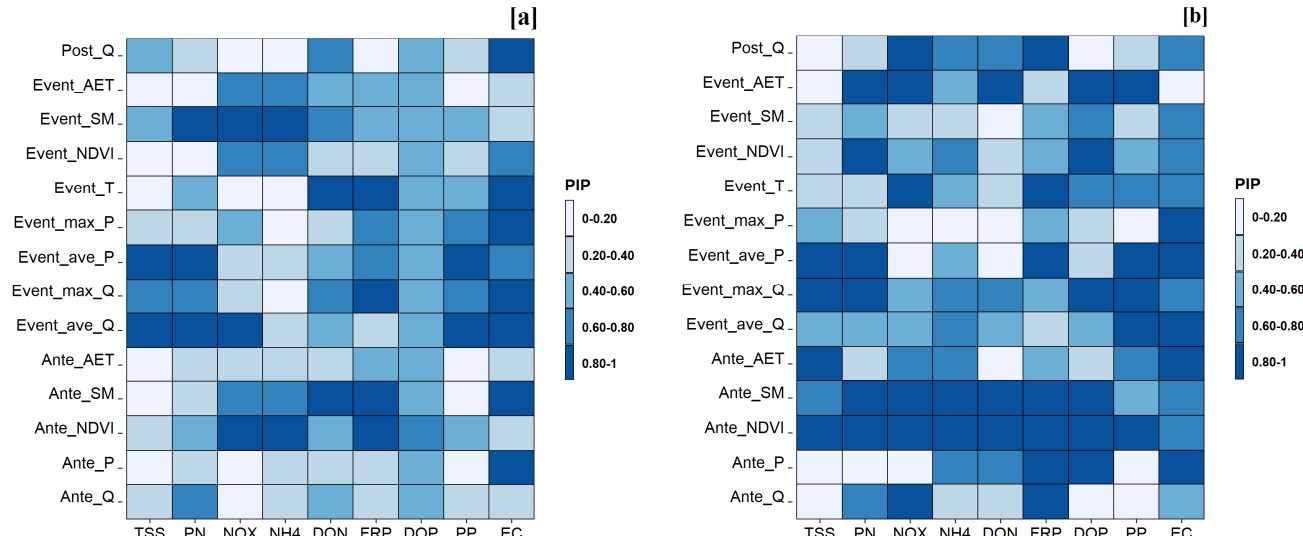

**Figure 4: Posterior inclusion probability (PIP) of each candidate predictor for [a] Cluster 1 ("wet") catchments, and [b] Cluster 2 ("dry") catchments; dark blue = high PIP; light blue = low PIP. The definition of the abbreviations of each predictor on the y-axis are in Table 3.**

Results from here on will focus mainly on TSS, $NO_X$ and FRP, due to their impacts on the marine receiving environment. Results for the other six constituents are in the Supplementary Materials. Figure 5 shows the posterior model probabilities for TSS, $NO_x$ and FRP for the 100 models with highest PMP (Figures B2 and B3 in Appendix B show other constituents). Red indicates a negative influence and blue a positive influence. The difference in PIP between the two clusters resulted in quite different plausible model structures (models with relatively high posterior model probability). A stand-out difference between the results for the two Clusters was antecedent vegetation cover condition (*Ante_NDVI*), which tended to be a more important predictor of TSS for Cluster two, than for Cluster one (Figure 5 [a]). In addition, the plausible models for Cluster 2 were generally more complex (with a larger number of predictors), expect for DOP and EC (Figures B2 and B3).






**Figure 5: Comparison of BMA model coefficients and cumulative model probabilities (only the first 100 models ranked according to the highest probability are shown) between Cluster 1 ("wet" - left) and Cluster 2 ("dry" - right) sites for [a] TSS, [b] NOx and [c] FRP. The order of predictors on the y-axis was ranked based on the posterior inclusion probability. Each column in the heatmap represents the one specific model (ranked from highest model probability from left to right) and the width of the column**





**is normalised by the posterior model probability (i.e., the widest columns indicate models with the largest increase in probability compared to the next most probable model). The colour indicates the direction of the coefficients: red = negative; blue = positive. The coefficient value was averaged across the posterior median value of the site-specific coefficient within each cluster (effect size, $\theta_{n,j}$, in Equation 6); the definition of the abbreviations of each predictor on the y-axis are in Table 3.**

The distribution of posterior model coefficients for the key predictors (Figure 6, Figures B4 and B5) further demonstrated that the key drivers of temporal variability in water quality vary between catchments and between constituents. During-event runoff and rainfall tended to have a positive effect on sediment and particulate constituents and, a negative effect on $NO_X$ and EC. In addition, there was strong negative effect of antecedent vegetation condition on the majority of the constituents.





**Figure 6: Distribution of median of site-level coefficients for all plausible models in BMA between Cluster 1 ("wet" - left) and Cluster 2 ("dry" - right) sites for: [a] TSS; [b] NOₓ and [c] FRP. Only predictors with PIP > 0.8 are included. For each specific model structure, the coefficient value of a predictor was the median of the site-specific coefficient across all sites (effect size, $\theta_{n,j}$, in Eq. (6)). The distribution of this value thus represents the probability of the model (PMP), as well as variability in the same predictor across different sites; black dots = the median; grey vertical lines = 95% CI; blue coloured vertical lines = 50% CI; the definition of the abbreviation of each predictor on x-axis are in Table 3.**

The uncertainty in PIP, derived from 1,000 subsampled BMA runs (Figure 7, Figures B6 and B7) highlighted that the BMA results were robust for most constituents, except for EC (Figure B7 [c]). BMA tends to identify important predictors and less sensitive to the input data which is evidenced by the relatively narrow range of interquartile ranges (IQR), when PIP for a specific predictors is large (e.g., antecedent soil moisture for FRP in Figure 7). It is also worth noting that large uncertainty in the PIP for EC was observed, indicating the BMA results were sensitive to the observations of EC. This might be related to data availability, which is further discussed in Sect. 4.2.

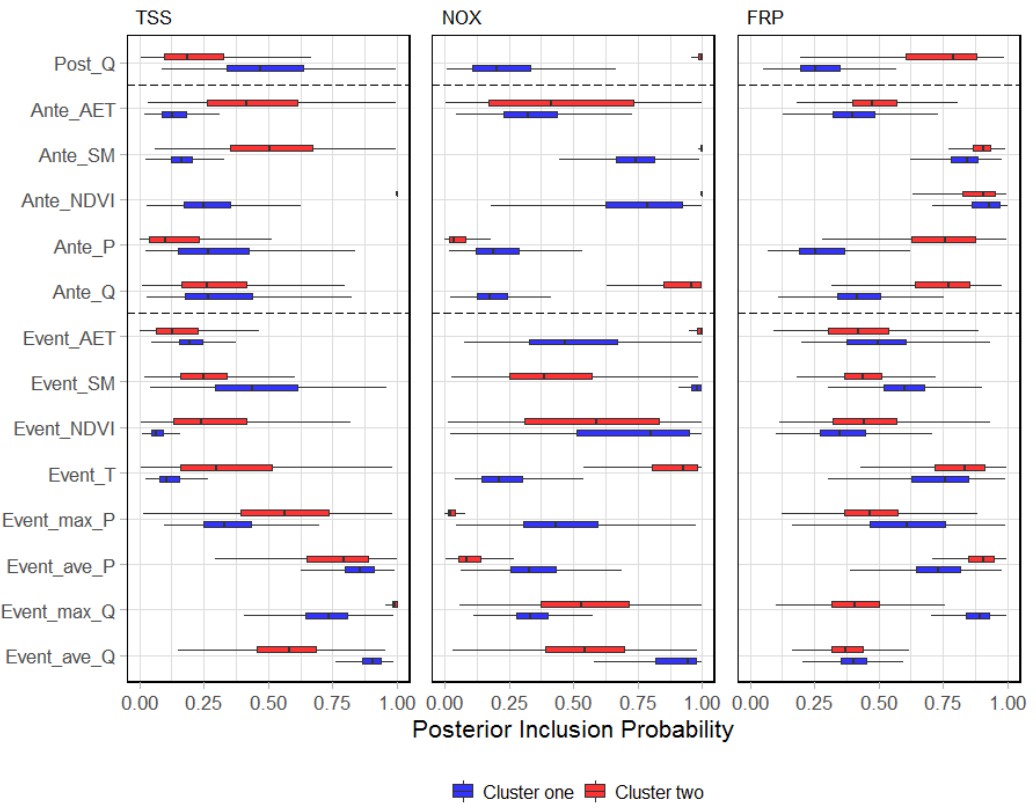

**Figure 7: The comparisons of the distribution of posterior inclusion probabilities of the individual predictors derived from 1,000 subsampled BMA runs; the boxes are the interquartile ranges (IQR, 25th to 75th percentile), and the whiskers are the ranges between 1.5 IQR of the lower quartile and 1.5 IQR of the higher quartile; the vertical bar = median; blue = Cluster 1 ("wet"); red = Cluster 2 ("dry"); the definition of abbreviation of each predictor on y-axis are in Table 3.**





## 3.2 Predictive performance

Moderate levels of temporal variability were explained by the BMA framework for the two independent clusters of sites
(Figure 8, Figures B8 and B9). At the cluster level, the NSE ranged from 0.04 (DOP) to 0.68 (EC) and from 0.34 ($NH_4$) to
0.64 ($NO_X$) for Clusters 1 and 2 (full model columns in Table 4), respectively. The comparison of the modelling
performance (posterior median of BMA prediction) showed that the modelling framework performed better on the Cluster 2
sites than Cluster 1 (Figure 8, red 50% prediction CI – Cluster 2), except for $NH_4$ and EC (not shown). This was reflected in
a better match to the 1:1 line within the 90% prediction CI for Cluster two catchments. It is also worth noting that the
prediction interval for EC (Figure B9 [c]) was much wider than the rest of the constituents. Similar results were found in the
site-level performance, with the average site-level NSE (Figure 9) for Cluster 2 models typical higher than for Cluster 1. The
site-specific performance varied across sites, with the largest variation in EC (NSE for the Cluster 2 result ranged from
approximately 0.20 to 0.90). The modelling performance of DOP in the Cluster 1 sites was poor (NSE = 0.04); all candidate
covariates had low predictive power, resulting in the poor mixing of chains of the inclusion variable $I_n$ (i.e., posterior $I_n$ was
around 0.5). The model residuals were normally distributed (Figure B10) and there was no clear heteroscedasticity within the
residuals (Figures B11 to B19).



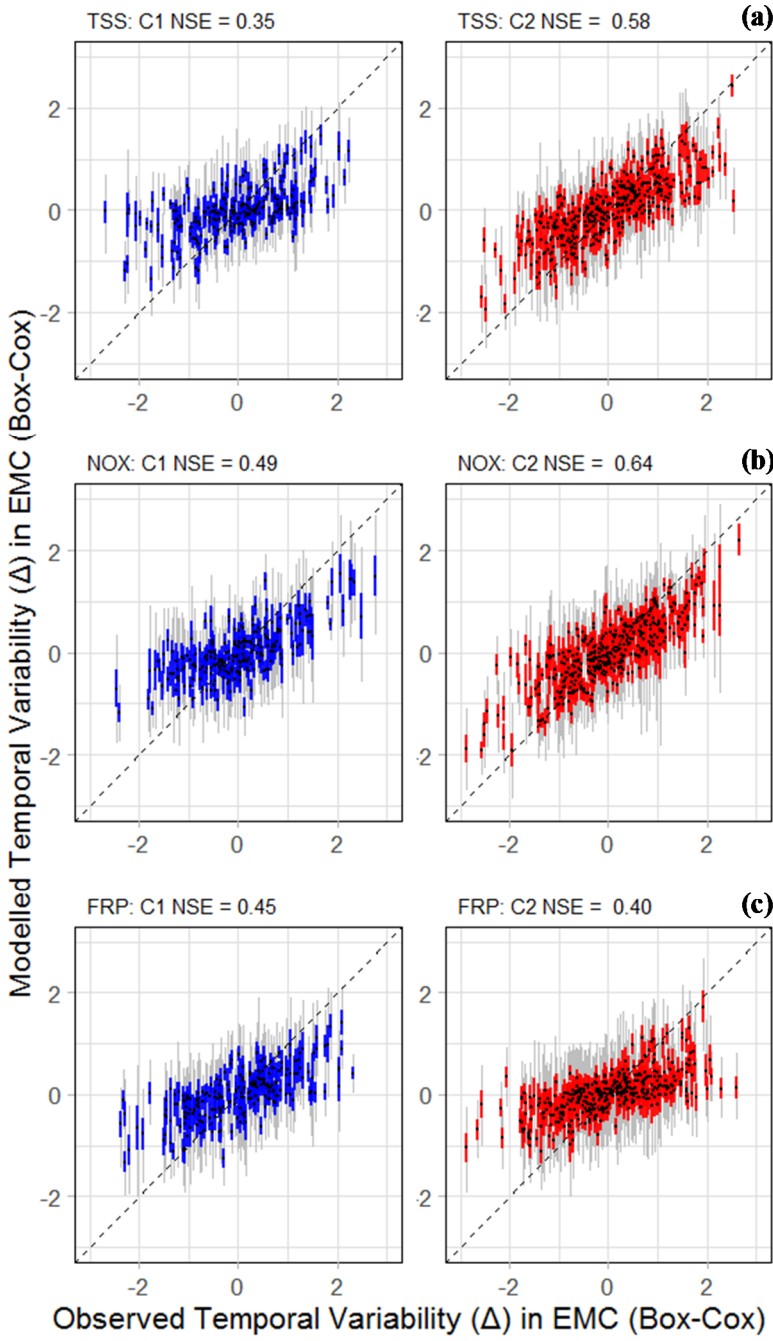

**Figure 8: Performance of the BMA models of the temporal variability of three constituents across 32 sites, represented by prediction intervals from BMA and observed Box-Cox EMC across two clusters of sites for: (a) TSS; (b) NOₓ; and (c) FRP. Each bar shows a single event and all events at all sites in the cluster are included. The NSE values were calculated based on median predictions. Black dots show prediction median; grey vertical lines show 95% CI; coloured vertical lines show 50% CI; blue is Cluster 1 ("wet"); red is Cluster 2 ("dry"); and dashed black lines are the 1:1 relationship.**





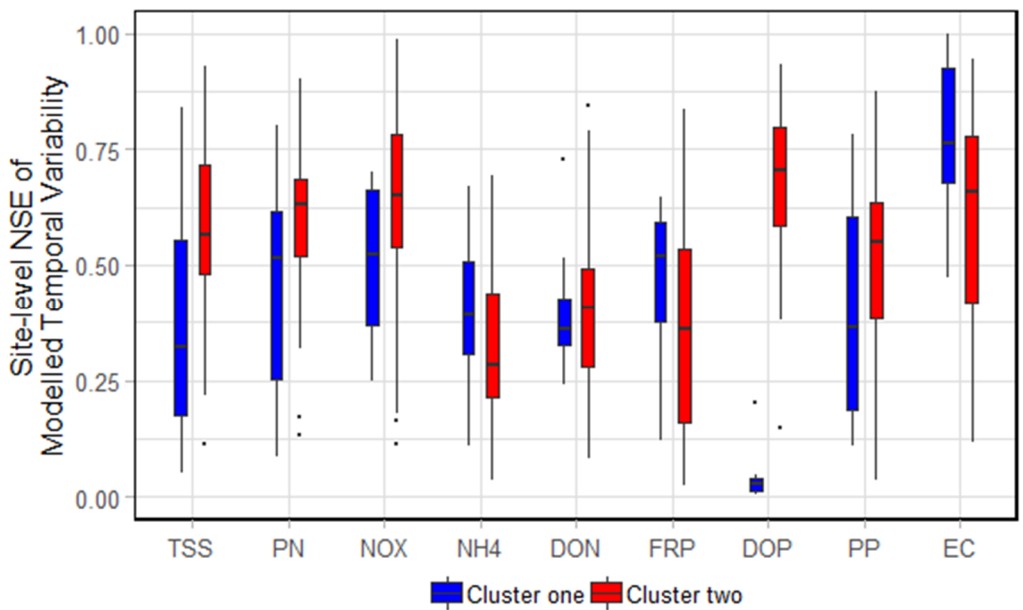

**Figure 9: Distribution of site-level NSE for modelled the temporal variability of two clusters of sites. The interpretation of boxplot**
**is the same as Figure 7. NSE values were calculated based on site-level predictions of event median EMC; blue is Cluster 1("wet");**
**and red is Cluster 2 ("dry") (i.e., each boxplot is comprised of respective number of sites in each cluster, one for each catchment).**

Table 4 compares the model performance using rainfall/runoff related predictors only and all candidate predictors (full

model). A large increase in NSE was found for most dissolved nutrient species (e.g., $NO_X$, $NH_4$, DON, FRP and DOP) for

the full model. Notably, for $NH_4$ in Cluster 1, factors other than rainfall and runoff explained almost all the variability that

could be captured by the BMA.

**Table 4: Comparison between BMA performance using rainfall/runoff predictors only and all candidate predictors (full models).**

| Constituent | NSE for Cluster 1 ("wet") | | | NSE for Cluster 2 ("dry") | | |
|---|---|---|---|---|---|---|
| | Rainfall, runoff only | Full model | % change in NSE | Rainfall, runoff only | Full model | % change in NSE |
| TSS | 0.32 | 0.35 | 11 | 0.42 | 0.58 | 38 |
| PN | 0.32 | 0.40 | 24 | 0.38 | 0.59 | 56 |
| $NO_X$ | 0.23 | 0.49 | 113 | 0.32 | 0.64 | 101 |
| $NH_4$ | 0.00 | 0.39 | / | 0.18 | 0.34 | 88 |
| DON | 0.20 | 0.37 | 84 | 0.20 | 0.43 | 117 |
| FRP | 0.27 | 0.45 | 68 | 0.26 | 0.40 | 56 |
| DOP | 0.00 | 0.04 | / | 0.22 | 0.62 | 181 |
| PP | 0.29 | 0.36 | 24 | 0.34 | 0.51 | 51 |
| EC | 0.41 | 0.68 | 66 | 0.39 | 0.54 | 39 |



## 4 Discussion

### 4.1 Factors influencing temporal variability in stream water quality

#### 4.1.1 Runoff and rainfall

Our results demonstrated that runoff and rainfall were important factors in explaining the temporal dynamics of particulate pollutants (i.e. TSS, PN and PP) and dissolved species (e.g., NOx, DOP and EC) in the GBR catchments. These results align with the findings of previous studies that have used these variables to understand changes in water quality over time (Beiter et al., 2020; Letcher et al., 2002; Liu et al., 2008b; McKergow et al., 2003; Schwarz et al., 2006; Tilburg et al., 2015). Hydrologic and climatic variables (i.e. rainfall and runoff) showed distinct effects on different constituents, as well as different groups of catchments. The positive effect of event runoff and rainfall on sediment and particulate nutrients (i.e., PN, PP) revealed their underlying impacts on pollutant mobilisation and transport processes in catchments (Ballantine et al., 2009; Guo et al., 2010; Heathwaite et al., 2000; Hirsch et al., 2010; Lintern et al., 2018b; Musolff et al., 2015). In contrast, there were negative effects of during-event runoff on NOx (Cluster 1), DOP (Cluster 2) and EC (both clusters). For NOx and EC, this was most likely caused by hydrological transport processes; these constituents tend to be transported to receiving rivers via subsurface flows (Kratz et al., 1997; McKergow et al., 2003). For events with relatively low surface runoff, higher NOx and EC event concentrations could be expected in these catchments (Clow and Sueker, 2000; Skoulikidis et al., 2006; Young et al., 1996). In addition, for DOP, in-stream biogeochemical cycling was likely to have caused the negative effect of event runoff. The events with low runoff, coupled with high temperatures (positive effect of event temperature for DOP Cluster 2, Figure B3 [a]) may relate to increases in the rate of P releases from organic forms at higher temperatures (Verheyen et al., 2015).

Post-event runoff (*Post_Q*) showed effects on specific constituents (e.g., NOx, FRP and EC). Two alternative reasons might explain this. First, high post-event runoff may be an indicator of large baseflow contribution during the events (Cuomo and Guida, 2016). Therefore, as discussed in the above paragraph, constituents that can be transported through subsurface flows tend to be influenced by amount of runoff after event. Alternatively, it was significantly and positively correlated with other event characteristics and catchment biophysical conditions (e.g., vegetation cover, Figure B1). These inter-correlated factors together could have influenced pollutant source, mobilisation and delivery (see discussions below) (Granger et al., 2010; Lintern et al., 2018a).

#### 4.1.2 Vegetation cover

Vegetation cover was another driving factor that was found to have influenced water quality dynamics; antecedent NDVI (*Ante_NDVI*) was included in the plausible models more frequently than event NDVI. The negative effect of antecedent





NDVI on particulate and dissolved nutrients (except DOP) was in line with previous studies that have found that NDVI was negatively correlated with these constituent concentrations in streams (Griffith et al., 2002; Liu et al., 2015; Masocha et al.,

2017). An explanation for these results could be that high vegetation groundcover tended to stabilise the surface soil and reduce sediment losses by erosion (Meyer et al., 1997; Singh et al., 2008). In addition, vegetation nutrient assimilation and retention processes consumed nutrients in sediment and waterbodies, and these processes peaked in spring and early summer, typically before the wet season in the GBR catchments (Tabacchi et al., 2000; Uwimana et al., 2018; Vymazal, 2007).

The effect of antecedent NDVI varied among groups of constituents in Clusters 1 and 2. Specifically, it was a key predictor for $NO_X$, $NH_4$ and FRP for Cluster one, and almost all constituents for Cluster 2. This can be explained by the contrasting landscapes and climate of these two regions (Liu et al., 2018). In the dense, vegetation-covered catchments in Cluster 1 (i.e., the sites in the Wet Tropics), dissolved inorganic nutrient losses were likely due to more fertile soils (e.g., application of fertiliser on sugarcane) during the growing season (Hunter and Walton, 2008; McKergow et al., 2005a). Furthermore, denser

natural vegetation cover (e.g., riparian vegetation and forest) could increase plant uptake and assimilation of dissolved nutrients compared to the sparse vegetation cover in the Dry Tropics (Cluster 2) region. Conversely, among Cluster 2 sites, vegetation coverage showed clear seasonal variation, which was linked closely to the seasonality in rainfall and grazing activity. Sediments and particulate pollutants were likely to be mobilized in grazed catchments (high rate of soil erosion) and delivered to streams via surface runoff (Ballantine et al., 2009; Neil et al., 2002; Turner et al., 2012). More importantly, high

vegetation cover tended to mitigate mobilisation of pollutants, through stabilising the surface soil and such that reduces sediment losses from erosion (Meyer et al., 1997; Rey, 2004; Singh et al., 2008; Zorzal-Almeida et al., 2018).

### 4.1.3 Soil moisture and evapotranspiration

The results showed that soil moisture (SM) and actual evapotranspiration (AET) had a high impact on different constituents, particularly in the Cluster 2 catchments (e.g., antecedent soil moisture [DON and EC], antecedent AET [TSS and EC]).

These two variables were inter-correlated and affect the hydrological cycle and vegetation cover (Correll, 1996; Correll and Weller, 1989; Legates et al., 2011). The results indicated that antecedent soil moisture had a negative effect on PN, $NO_X$, $NH_4$, DON, DOP and FRP. On one hand, this was expected as antecedent soil moisture was positively correlated with vegetation cover, and high soil moisture tends to reduce soil erosion and increase plant nutrient uptake. It may also be that soil water content affected soil microbial activity, influencing the biogeochemical processes in catchments, such as

denitrification (Doran et al., 1988; Doran, 1980; Weier et al., 1993). The rate of denitrification was also enhanced under anoxic conditions, when soil moisture was high (Skopp et al., 1990; Zhu et al., 2018a). On the other hand, higher soil water can be associated with increased shallow subsurface flow and leaching of some constituents such as $NO_X$ (Zhu et al., 2018b). This appears not to occur to a sufficient extent for it to over-ride other impacts of soil moisture.



### 4.1.4 Temperature

Our results suggested that average event temperature (*Event_T*) had a positive effect on $NO_X$, FRP, and DOP. This may be attributed to the strong negative cross-correlation between temperature and event runoff and antecedent vegetation condition (Figure B1). Rainfall during a warmer period might have been associated with less event runoff, resulting in higher event mean concentrations (Sect. 4.1.1). The effect of event temperature can be also attributed to the fact that the higher temperatures could lead to more recent mineralisation of nutrients, increasing readily transportable dissolved nutrient sources

(Liu et al., 2017; Wang et al., 2020). In addition, higher event temperature might be associated with higher pre-event temperature, resulting in poor groundcover, potentially lowering the dissolved nutrients losses through plant assimilation/uptake (Sect. 0) (Muro et al., 2018).

### 4.2 Predicting temporal variations in water quality

The Bayesian modelling framework in this study provided a useful tool to assess in-stream water quality dynamics. The

models were able to explain more temporal variation in $NO_X$ and EC than in other constituents. This is related to the sources and delivery processes of these two constituents. Anthropogenic inputs (e.g. agriculture) for $NO_X$, and large stores in groundwater together with limited geochemical transformation for EC (salts) suggested that temporal changes in event concentration could be well-captured by the changes in catchment hydroclimatic and vegetation conditions. In addition, $NO_X$ and EC tend to be transported in subsurface flow pathways. The dynamics of catchment soil wetness and vegetation cover

have been previously linked to hydrological interactions between surface and subsurface flows (Ursino et al., 2004). The incorporation of soil moisture and vegetation cover into the Bayesian modelling framework more readily allowed the description of the main ecohydrological processes of these two constituents.

In contrast, model performance for DOP was poor in Cluster 1 catchments, which can be explained by two reasons. First, in the Wet Tropics catchments, DOP concentrations were generally stable, regardless of changes in flow, which can be

explained by chemical exchange processes between water and sediment in stream (White et al., 1998). This means that the variability in DOP cannot be captured by the environmental variables considered here. Second, the poor performance might be attributed to the data set having fewer observations of DOP EMCs among Cluster 1 sites. There were only 66 observations, compared to the next lowest number of 167 (EC) among other constituents in the Cluster 1 catchments, which may not be sufficient to fully inform the model. This small sample size could have led to outcomes of: 1) poor mixing of

MCMC chains for inclusion variables (Figure B7 [a]), where no predictors showed predictive power; and 2) the BMA failed to identify the plausible models, since none of the candidate models had enough predictive power to fit the data well (Guthke, 2017; Höge et al., 2019). Continuous DOP monitoring would be required to achieve a better understanding of the factors driving temporal variation in this constituent.

Statistical modelling in hydrology or water quality is affected by uncertainty, only some of which can be characterised

within any particular modelling framework (Beck, 1987; Kavetski et al., 2006; Mantovan et al., 2006; Renard et al., 2010;





Yang et al., 2007). The Bayesian modelling framework used in this study incorporated the uncertainties in model selection (between-model), observations and model parameters (within-model) directly into the model predictions (Steel, 2019). This is a more comprehensive characterisation than in studies where model structures are assumed a priori. Reporting of predictive uncertainty of temporal variations in water quality also provided valuable information on the confidence in the

averaged predictions. Nevertheless, limitations remain in the BMA approach which are important to understand. For example, for EC, there was a larger predictive uncertainty and larger uncertainty in posterior inclusion probability for each predictor from the robustness assessment than estimated in the fit to the complete data set. One limitation of BMA is that the posterior model probability could be sensitive to the specification of the parameter prior distribution (Fernandez et al., 2001). Specifying more informative priors on model parameters (i.e., inclusion variable $I_n$) would have the effect of restricting the

set of candidate models (Eicher et al., 2011; Rockey et al., 2016). Indeed, several studies have compared different predictive performances of different prior specification of BMA coefficients and found that the choice of prior matters (Bayarri et al., 2012; Eicher et al., 2011; Liang et al., 2008). Future investigation of the sensitivity of prior distributions for BMA coefficients might achieve a reduction in predictive uncertainty and instability in posterior inclusion probabilities.

**4.3 Management implications**

The identification of key drivers of temporal variation in water quality can inform catchment water quality management. The results of this study showed that the effects of hydro-climatic drivers (e.g., rainfall and runoff) and vegetation cover varied among constituents and regions. This may allow funding bodies, such as government, regional natural resource management groups, to identify regions where land management and restoration would have a greater effect on mitigating sediments and nutrients export. The results suggested that, compared to wet catchments, maintaining vegetation ground

cover in large dry grazed catchments (e.g., the Burdekin and Fitzroy catchments in Cluster 2) before the wet seasons could be an effective way of reducing sediment losses via erosion processes. These results are consistent with current, improved land management practices across the GBR catchments (Brodie et al., 2009; Brodie et al., 2012; Government, 2017; Hunter and Walton, 2008; Star et al., 2015). Management measures (e.g., establishment of wetlands, re-vegetation/rehabilitation of gully and stabilisation of river banks) can reduce sediment losses from hillslope and gully erosions (Koci et al., 2020; Loch,

2000; Sherriff et al., 2016; Wilkinson et al., 2018). In addition, catchment-specific management that accounts for temporal variation in catchment hydrological connectivity is required for the control of dissolved nutrients. Dominant flow pathways for dissolved nutrients can vary spatially and temporally. For example, subsurface flow in the Wet Tropics region have tended to transmit more dissolved nutrient, because prolonged wet conditions lead to this region that is more likely to be connected via lateral subsurface flow (Geng et al., 2017; Stieglitz et al., 2003). The enhanced mobilisation of leached

dissolved nutrients from intensive cropping (e.g., sugarcane) from perched groundwater should be targeted in these catchments (Melland et al., 2012). Management practices, such as conservation tillage, and adaptation of '4R'concept (right source, right rate, right time, right place) for fertiliser application may help to minimise dissolved nitrogen losses (Cestti et al., 2003; Lintern et al., 2020; Merriman et al., 2009; Snyder, 2017).





## 5. Conclusions

This study provides a data-driven understanding of key drivers influencing the temporal variation in water quality. A hierarchical Bayesian model averaging framework was used to identify the key environmental drivers and predict the water quality dynamics at multiple catchments. Results showed that the temporal dynamics of water quality can be predicted well using models considering the combined effects of hydroclimate and vegetation groundcover. The effects of key hydro-climatic and vegetation conditions varied among different constituents, and across regions. This study reinforces the

importance of vegetation cover management as one key management response, especially for large grazed catchments. Future investigation could involve the development of a spatio-temporal modelling framework to fully capture the water quality dynamics. More importantly, it has continued to be challenging to prioritise management practices and evaluate the effectiveness of the improved management interventions. Consequently, with more land management surveys and continuous water quality monitoring data available, an extended temporal or spatio-temporal modelling framework could

potentially be used to assess if the success of the restoration measures.

### Data availability

Water quality data that supported this study was available upon request from the Great Barrier Reef Catchment Loads Monitoring Program (GBREvents@dsiti.qld.gov.au). Sources of explanatory variables were listed in Table 2.

### Author contribution

All authors contributed to the design of the research. SL carried out data collation, performed the simulations and prepared the manuscript with contributions from all co-authors. All authors contributed to the interpretation of the results and provided feedback.

### Competing interests

The authors declare that they have no conflict of interest.

### Acknowledgements

This study was supported by the Australian Research Council (LP140100495), the Environment Protection Authority Victoria, the Victorian Department of Environment, Land, Water and Planning, Bureau of Meteorology and Queensland Department of Natural Resources, Mines and Energy. The author would like to acknowledge the efforts of the Queensland Department of Environment and Science who provided the water quality monitoring data. The authors would also like to





offer sincere gratitude to Ms. Jie Jian for her assistance in geospatial database compilation. Dr. Paul Leahy, Mr. Malcolm Watson, Dr. Ulrike Bende-Michl, Mr. Paul Wilson, and Ms. Belinda Thompson all of whom provided valuable advice in the preparation of this manuscript.

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

Lloyd, C., Freer, J., Johnes, P., and Collins, A.: Using hysteresis analysis of high-resolution water quality monitoring data, including uncertainty, to infer controls on nutrient and sediment transfer in catchments, Science of the Total Environment, 543, 388-404, 2016.

Loch, R.: Effects of vegetation cover on runoff and erosion under simulated rain and overland flow on a rehabilitated site on the Meandu Mine, Tarong, Queensland, Soil Research, 38, 299-312, 2000.

Mainali, J. and Chang, H.: Landscape and anthropogenic factors affecting spatial patterns of water quality trends in a large river basin, South Korea, Journal of hydrology, 564, 26-40, 2018.

Mainali, J., Chang, H., and Chun, Y.: A review of spatial statistical approaches to modeling water quality, Progress in Physical Geography: Earth and Environment, 0, 0309133319852003, 2019.

Masocha, M., Murwira, A., Magadza, C. H., Hirji, R., and Dube, T.: Remote sensing of surface water quality in relation to catchment condition in Zimbabwe, Physics and Chemistry of the Earth, Parts A/B/C, 100, 13-18, 2017.

McGrane, S. J.: Impacts of urbanisation on hydrological and water quality dynamics, and urban water management: a review, Hydrological Sciences Journal, 61, 2295-2311, 2016.

McKergow, L. A., Prosser, I. P., Hughes, A. O., and Brodie, J.: Regional scale nutrient modelling: exports to the Great Barrier Reef world heritage area, Marine pollution bulletin, 51, 186-199, 2005a.

McKergow, L. A., Prosser, I. P., Hughes, A. O., and Brodie, J.: Sources of sediment to the Great Barrier Reef world heritage area, Marine pollution bulletin, 51, 200-211, 2005b.

McKergow, L. A., Weaver, D. M., Prosser, I. P., Grayson, R. B., and Reed, A. E.: Before and after riparian management: sediment and nutrient exports from a small agricultural catchment, Western Australia, Journal of Hydrology, 270, 253-272, 2003.

Melland, A., Mellander, P.-E., Murphy, P., Wall, D., Mechan, S., Shine, O., Shortle, G., and Jordan, P.: Stream water quality in intensive cereal cropping catchments with regulated nutrient management, Environmental science & policy, 24, 58-70, 2012.

Merriman, K. R., Gitau, M., and Chaubey, I.: A tool for estimating best management practice effectiveness in Arkansas, Applied Engineering in Agriculture, 25, 199-213, 2009.

Meyer, D. L., Townsend, E. C., and Thayer, G. W.: Stabilization and erosion control value of oyster cultch for intertidal marsh, Restoration Ecology, 5, 93-99, 1997.

Miller, C., Magdalina, A., Willows, R., Bowman, A., Scott, E., Lee, D., Burgess, C., Pope, L., Pannullo, F., and Haggarty, R.: Spatiotemporal statistical modelling of long-term change in river nutrient concentrations in England & Wales, Science of the Total Environment, 466, 914-923, 2014.

Muro, J., Strauch, A., Heinemann, S., Steinbach, S., Thonfeld, F., Waske, B., and Diekkrüger, B.: Land surface temperature trends as indicator of land use changes in wetlands, International journal of applied earth observation and geoinformation, 70, 62-71, 2018.

Murphy, B. F. and Ribbe, J.: Variability of southeastern Queensland rainfall and climate indices, International Journal of Climatology: A Journal of the Royal Meteorological Society, 24, 703-721, 2004.

Musolff, A., Schmidt, C., Selle, B., and Fleckenstein, J. H.: Catchment controls on solute export, Advances in Water Resources, 86, 133-146, 2015.

Nash, J. E. and Sutcliffe, J. V.: River flow forecasting through conceptual models part I—A discussion of principles, Journal of hydrology, 10, 282-290, 1970.

Neil, D. T., Orpin, A. R., Ridd, P. V., and Yu, B.: Sediment yield and impacts from river catchments to the Great Barrier Reef lagoon: a review, Marine and Freshwater Research, 53, 733-752, 2002.

Noori, N., Kalin, L., and Isik, S.: Water quality prediction using SWAT-ANN coupled approach, Journal of Hydrology, 590, 125220, 2020.

Novotny, V.: Diffuse pollution from agriculture—a worldwide outlook, Water Science and Technology, 39, 1-13, 1999.

Ntzoufras, I.: Gibbs variable selection using BUGS, Journal of statistical software, 7, 1-19, 2002.

O'Hara, R. B. and Sillanpää, M. J.: A review of Bayesian variable selection methods: what, how and which, Bayesian analysis, 4, 85-117, 2009.

Orr, D., Turner, R. D. R., Huggins, R., Vardy, S., and J., W. M. S.: Wet Tropics water quality statistics for high and base flow conditions., Great Barrier Reef Catchment Loads Monitoring Program, D. o. S., Information and Technology, I. a. t. A. (Eds.), Brisbane, 2014.

Paliwal, R., Sharma, P., and Kansal, A.: Water quality modelling of the river Yamuna (India) using QUAL2E-UNCAS, Journal of environmental management, 83, 131-144, 2007.

Papagiannopoulou, C., Miralles, D., Dorigo, W. A., Verhoest, N., Depoorter, M., and Waegeman, W.: Vegetation anomalies caused by antecedent precipitation in most of the world, Environmental Research Letters, 12, 074016, 2017.

Peel, M. C., Finlayson, B. L., and McMahon, T. A.: Updated world map of the Köppen-Geiger climate classification, Hydrology and earth system sciences discussions, 4, 439-473, 2007.

Pérez-Gutiérrez, J. D., Paz, J. O., and Tagert, M. L. M.: Seasonal water quality changes in on-farm water storage systems in a south-central US agricultural watershed, Agricultural water management, 187, 131-139, 2017.

Peters, N. E. and Meybeck, M.: Water quality degradation effects on freshwater availability: impacts of human activities, Water International, 25, 185-193, 2000.

Pilgrim, E., Institution of Engineers, A., Pilgrim, D., and Canterford, R.: Australian rainfall and runoff, Institution of Engineers, Australia, 1987.

Plummer, M.: JAGS: A program for analysis of Bayesian graphical models using Gibbs sampling, 2003.

Plummer, M.: JAGS: Just another Gibbs sampler, version 3.4. 0, URL http://mcmc-jags. sourceforge. net, 2013a. 2013a.

Plummer, M.: rjags: Bayesian graphical models using MCMC, R package version, 3, 2013b.

Pretty, J., Hildrew, A., and Trimmer, M.: Nutrient dynamics in relation to surface–subsurface hydrological exchange in a groundwater fed chalk stream, Journal of Hydrology, 330, 84-100, 2006.

R Core Team: R: A language and environment for statistical computing, 2013. 2013.

Raftery, A. E., Madigan, D., and Hoeting, J. A.: Bayesian model averaging for linear regression models, Journal of the American Statistical Association, 92, 179-191, 1997.

Raupach, M., Briggs, P., Haverd, V., King, E., Paget, M., and Trudinger, C.: Australian water availability project (AWAP): CSIRO marine and atmospheric research component: final report for phase 3, Melbourne: Centre for Australian weather and climate research (bureau of meteorology and CSIRO), 67, 2009.

Ren, W., Zhong, Y., Meligrana, J., Anderson, B., Watt, W. E., Chen, J., and Leung, H.-L.: Urbanization, land use, and water quality in Shanghai: 1947–1996, Environment International, 29, 649-659, 2003.

Rey, F.: Effectiveness of vegetation barriers for marly sediment trapping, Earth Surface Processes and Landforms: The Journal of the British Geomorphological Research Group, 29, 1161-1169, 2004.

Richards, R. P. and Baker, D. B.: Trends in nutrient and suspended sediment concentrations in Lake Erie tributaries, 1975–1990, Journal of Great Lakes Research, 19, 200-211, 1993.

Rode, M., Arhonditsis, G., Balin, D., Kebede, T., Krysanova, V., Van Griensven, A., and Van der Zee, S. E.: New challenges in integrated
water quality modelling, Hydrological Processes, 24, 3447-3461, 2010.

Schwarz, G., Hoos, A., Alexander, R., and Smith, R.: The SPARROW surface water-quality model: theory, application and user documentation, US geological survey techniques and methods report, book, 6, 2006.

Sharpley, A.: Managing agricultural phosphorus to minimize water quality impacts, Scientia Agricola, 73, 1-8, 2016.

Sherriff, S., Rowan, J., Melland, A., Jordan, P., Fenton, O., and O hUallachain, D.: Investigating suspended sediment dynamics in
contrasting agricultural catchments using ex situ turbidity-based suspended sediment monitoring, Hydrology and Earth System Sciences, 19, 3349-3363, 2015.

Sherriff, S. C., Rowan, J. S., Fenton, O., Jordan, P., Melland, A. R., Mellander, P.-E., and Huallachain, D. O.: Storm event suspended sediment-discharge hysteresis and controls in agricultural watersheds: implications for watershed scale sediment management, Environmental science & technology, 50, 1769-1778, 2016.

Shi, P., Zhang, Y., Li, Z., Li, P., and Xu, G.: Influence of land use and land cover patterns on seasonal water quality at multi-spatial scales, Catena, 151, 182-190, 2017.

Singh, A., Jakubowski, A. R., Chidister, I., and Townsend, P. A.: A MODIS approach to predicting stream water quality in Wisconsin, Remote Sensing of Environment, 128, 74-86, 2013.

Singh, P., Bhunya, P., Mishra, S., and Chaube, U.: A sediment graph model based on SCS-CN method, Journal of Hydrology, 349, 244-
750  255, 2008.

Skopp, J., Jawson, M., and Doran, J.: Steady-state aerobic microbial activity as a function of soil water content, Soil Science Society of America Journal, 54, 1619-1625, 1990.

Skoulikidis, N. T., Amaxidis, Y., Bertahas, I., Laschou, S., and Gritzalis, K.: Analysis of factors driving stream water composition and synthesis of management tools—a case study on small/medium Greek catchments, Science of the Total Environment, 362, 205-241, 2006.

Snyder, C. S.: Enhanced nitrogen fertiliser technologies support the '4R'concept to optimise crop production and minimise environmental losses, Soil Research, 55, 463-472, 2017.

Star, M., Rolfe, J., Long, P., Whish, G., and Donaghy, P.: Improved grazing management practices in the catchments of the Great Barrier Reef, Australia: does climate variability influence their adoption by landholders?, The Rangeland Journal, 37, 507-515, 2015.

Stieglitz, M., Shaman, J., McNamara, J., Engel, V., Shanley, J., and Kling, G. W.: An approach to understanding hydrologic connectivity
on the hillslope and the implications for nutrient transport, Global biogeochemical cycles, 17, 2003.

Tabacchi, E., Lambs, L., Guilloy, H., Planty-Tabacchi, A. M., Muller, E., and Decamps, H.: Impacts of riparian vegetation on hydrological processes, Hydrological processes, 14, 2959-2976, 2000.

Tang, W. and Carey, S. K.: HydRun: A MATLAB toolbox for rainfall–runoff analysis, Hydrological Processes, 31, 2670-2682, 2017.

Thompson, S., Basu, N., Lascurain, J., Aubeneau, A., and Rao, P.: Relative dominance of hydrologic versus biogeochemical factors on
solute export across impact gradients, Water Resources Research, 47, 2011.

Tilburg, C. E., Jordan, L. M., Carlson, A. E., Zeeman, S. I., and Yund, P. O.: The effects of precipitation, river discharge, land use and coastal circulation on water quality in coastal Maine, Royal Society open science, 2, 140429, 2015.

Turner, R., Huggins, R., Wallace, R., Smith, R., Vardy, S., and Warne, M. S. J.: Sediment, Nutrient and Pesticide Loads: Great Barrier Reef Catchment Loads Monitoring 2009-2010, Department of Science, Information Technology, Innovation and the Arts, Brisbane, 2012.
770  53, 2012.





Tweed, S. O., Leblanc, M., Webb, J. A., and Lubczynski, M. W.: Remote sensing and GIS for mapping groundwater recharge and discharge areas in salinity prone catchments, southeastern Australia, Hydrogeology Journal, 15, 75-96, 2007.

Ustaoğlu, F., Tepe, Y., and Taş, B.: Assessment of stream quality and health risk in a subtropical Turkey river system: A combined approach using statistical analysis and water quality index, Ecological Indicators, 113, 105815, 2020.

Uwimana, A., van Dam, A. A., Gettel, G. M., and Irvine, K.: Effects of agricultural land use on sediment and nutrient retention in valley-bottom wetlands of Migina catchment, southern Rwanda, Journal of environmental management, 219, 103-114, 2018.

Varanou, E., Gkouvatsou, E., Baltas, E., and Mimikou, M.: Quantity and quality integrated catchment modeling under climate change with use of soil and water assessment tool model, Journal of Hydrologic Engineering, 7, 228-244, 2002.

Verheyen, D., Van Gaelen, N., Ronchi, B., Batelaan, O., Struyf, E., Govers, G., Merckx, R., and Diels, J.: Dissolved phosphorus transport
from soil to surface water in catchments with different land use, Ambio, 44, 228-240, 2015.

Vymazal, J.: Removal of nutrients in various types of constructed wetlands, Science of the total environment, 380, 48-65, 2007.

Walling, D.: Dissolved loads and their measurement, Erosion and Sediment Yield: Some Methods of Measurement and Modelling. Geo Books, Regency House Norwich(England). 1984. p 111-177, 18 fig, 10 tab, 104 ref., 1984. 1984.

Walling, D. and Foster, I.: Variations in the natural chemical concentration of river water during flood flows, and the lag effect: some
further comments, Journal of Hydrology, 26, 237-244, 1975.

Wan, Y., Qian, Y., Migliaccio, K. W., Li, Y., and Conrad, C.: Linking spatial variations in water quality with water and land management using multivariate techniques, Journal of environmental quality, 43, 599-610, 2014.

Wang, A., Yang, D., and Tang, L.: Spatiotemporal variation in nitrogen loads and their impacts on river water quality in the upper Yangtze River basin, Journal of Hydrology, 590, 125487, 2020.

Wang, G., Jager, H. I., Baskaran, L. M., Baker, T. F., and Brandt, C. C.: SWAT modeling of water quantity and quality in the Tennessee river basin: spatiotemporal calibration and validation, Hydrology and Earth System Sciences Discussions, 2016. 1-33, 2016.

Wang, Q., Schepen, A., and Robertson, D. E.: Merging seasonal rainfall forecasts from multiple statistical models through Bayesian model averaging, Journal of Climate, 25, 5524-5537, 2012.

Waterhouse, J., Schaffelke, B., Bartley, R., Eberhadr, R., Brodie, J., Star, M., Thorburn, P., Rolfe, J., Ronan, M., Taylor, B., and Kroon,
F.: 2017 Scientific Consensus Statement:  A synthesis of the science of land-based water quality impacts on the Great Barrier Reef., Department of the Premier and Cabinet, Q. G., Brisbane (Ed.), Brisbane, 2017.

Waters, D., Carroll, C., Ellis, R., Hateley, L., McCloskey, J., Packett, R., Dougall, C., and Fentie, B.: Modelling reductions of pollutant loads due to improved management practices in the Great Barrier Reef Catchments-Whole of GBR, Volume 1 Department of Natural Resources and Mines, Technical Report (ISBN: 978-1-7423-0999), 2013.

Waters, D. and Packett, R.: Sediment and nutrient generation rates for Queensland rural catchments-an event monitoring program to improve water quality modelling, 2007.

Webb, A. and King, E. L.: A Bayesian hierarchical trend analysis finds strong evidence for large-scale temporal declines in stream ecological condition around Melbourne, Australia, Ecography, 32, 215-225, 2009.

Weier, K., Doran, J., Power, J., and Walters, D.: Denitrification and the dinitrogen/nitrous oxide ratio as affected by soil water, available
carbon, and nitrate, Soil Science Society of America Journal, 57, 66-72, 1993.

Whistler, J.: A phenological approach to land cover characterization using Landsat MSS data for analysis of nonpoint source pollution, KARS Rep, 96, 1-59, 1996.

Wilkinson, S. N., Kinsey-Henderson, A. E., Hawdon, A. A., Hairsine, P. B., Bartley, R., and Baker, B.: Grazing impacts on gully dynamics indicate approaches for gully erosion control in northeast Australia, Earth Surface Processes and Landforms, 43, 1711-1725,
810   2018.

Wintle, B. A., McCarthy, M. A., Volinsky, C. T., and Kavanagh, R. P.: The use of Bayesian model averaging to better represent uncertainty in ecological models, Conservation biology, 17, 1579-1590, 2003.

Yang, J.-L., Zhang, G.-L., Shi, X.-Z., Wang, H.-J., Cao, Z.-H., and Ritsema, C. J.: Dynamic changes of nitrogen and phosphorus losses in ephemeral runoff processes by typical storm events in Sichuan Basin, Southwest China, Soil and Tillage Research, 105, 292-299, 2009.

Young, W. J., Marston, F. M., and Davis, R. J.: Nutrient exports and land use in Australian catchments, Journal of environmental management, 47, 165-183, 1996.

Zhang, Q. and Blomquist, J. D.: Watershed export of fine sediment, organic carbon, and chlorophyll-a to Chesapeake Bay: spatial and temporal patterns in 1984–2016, Science of the Total Environment, 619, 1066-1078, 2018.

Zhang, Q., Harman, C. J., and Ball, W. P.: An improved method for interpretation of riverine concentration-discharge relationships
indicates long-term shifts in reservoir sediment trapping, Geophysical Research Letters, 43, 10,215-210,224, 2016.

Zhang, T. and Yang, B.: Box–cox transformation in big data, Technometrics, 59, 189-201, 2017.

Zhang, Y.-K. and Schilling, K.: Temporal variations and scaling of streamflow and baseflow and their nitrate-nitrogen concentrations and loads, Advances in Water Resources, 28, 701-710, 2005.

Zhang, Y., Guo, F., Meng, W., and Wang, X.-Q.: Water quality assessment and source identification of Daliao river basin using
multivariate statistical methods, Environmental monitoring and assessment, 152, 105, 2009.



Zhu, G., Wang, S., Wang, C., Zhou, L., Zhao, S., Li, Y., Li, F., Jetten, M. S., Lu, Y., and Schwark, L.: Resuscitation of anammox bacteria after> 10,000 years of dormancy, The ISME journal, 2018a. 2018a.

Zhu, Q., Castellano, M. J., and Yang, G.: Coupling soil water processes and nitrogen cycle across spatial scales: Potentials, bottlenecks and solutions, Earth-science reviews, 2018b. 2018b.

Zorzal-Almeida, S., Salim, A., Andrade, M. R. M., de Novaes Nascimento, M., Bini, L. M., and Bicudo, D. C.: Effects of land use and spatial processes in water and surface sediment of tropical reservoirs at local and regional scales, Science of the total environment, 644, 237-246, 2018.





**Appendix A - Text**

**Hierarchical prior specification and Bayesian inference of key drivers**

Bayesian inference required specification of prior distributions for each model parameter. A minimally-informative uniform prior (denote as $U(\cdot)$) between 0 and 10 was assigned to the global standard deviation ($\sigma$, Eq. A1) (Gelman, 2006). The prior of $I_n$ assumes that each indicator comes from an independent Bernoulli distribution, with a probability of 0.5 (Eq. A2) (Raftery et al., 1997). This vague prior results in each model structure having an equal prior model probability.

$$\sigma \sim U(0,10) \tag{A1}$$

$$I_n \sim Bernoulli(0.5) \tag{A2}$$

We used a hierarchical conditional prior specification for predictor coefficients, allowing the site-specific parameter values that describe the effects of each temporal predictors ($\beta_{1,j}, \beta_{2,j\ldots}, \beta_{n,j}$) to be exchangeable between sites (Liu et al., 2008; O'Hara and Sillanpää, 2009; Webb and King, 2009). The prior of $\beta_{n,j}$ was conditioned on $I_n$, resulting in a mixture distribution with 'slab and spike' prior, which was defined as follows,

$$\beta_{n,j}\big|I_n \sim I_n N(0,\tau_n) + (1-I_n)N(0,\tau_{n,tune}) \tag{A3}$$

where $\beta_{n,j} \mid (I_n = 1)$ is the slab part of the mixture distribution. The $\beta_{n,j} \mid (I_n = 1)$ was estimated by including a higher-level
distribution. The prior of $\beta_{n,j} \mid (I_n = 1)$ followed a normal distribution with random effect (Eq. A4), with the $\tau_n$ drawn from a common prior distribution, defined as a hyperparameter (i.e., uniform distribution between 0 to 20, Eq. A5) (Gelman, 2006; Kruschke, 2014).

$$\beta_{n,j}\big|(I_n=1) \sim N(0,\tau_n) \tag{A4}$$

$$\tau_n \sim U(0,20) \tag{A5}$$

For the spike component, a data-dependent prior was specified for $\beta_{n,j} \mid (I_n = 0)$, drawing from a *pseudo-prior* (Eq. A6), that is, *a prior* distribution with no effect on the posterior distribution, but facilitating the mixing of the Gibbs sampler.

$$\beta_{n,j}\big|(I_n=0) \sim N(0,\tau_{n,tune}) \tag{A6}$$

We estimated $\tau_{n,tune}$ from the standard deviations of the posterior of the $\beta_{n,j}$ in a global model structure (i.e., modelling structure using all predictors), as suggested by Carlin and Chib (1995) and Linden and Roloff (2015). The prior of $\beta_{n,j} \mid (I_n = 0)$ was near the posterior estimates to facilitate mixing in the MCMC (Hooten and Hobbs, 2015).

The posterior inclusion probability (PIP - $P(I_n =1\big|\boldsymbol{y})$, Eq.A7) of each predictor was used to compare the relative importance of individual predictors (i.e., how often the $n^{th}$ predictor was 'in' the model).

$$P(I_n =1\big|\boldsymbol{y}) = \frac{1}{T}\sum_{t=1}^{T} I(I_n^{(t)} = 1) \tag{A7}$$



where $T$ is the total number of iterations of Markov chains. The different combination of $I_n$ at each MCMC sampling represents a specific model structure. According to Bayes' theorem, the posterior model probability (PMP - $P(M_k|\mathbf{y})$) can be estimated as,

$$P(M_k | \mathbf{y}) = \frac{[\mathbf{y}|M_k]P(M_k)}{\sum_{x=1}^{L}[\mathbf{y}|M_x]P(M_x)} \qquad \text{A8}$$

where $L$ is the total number of possible models, and $P(M_k)$ is the prior probability of model $M_k$, among a group of models $M_x$, $x = 1, \ldots, X$. This posterior model probability can be obtained by assessing the frequency of a particular combination of $I_n$

during the MCMC sampling.

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



**Appendix B - Figure**

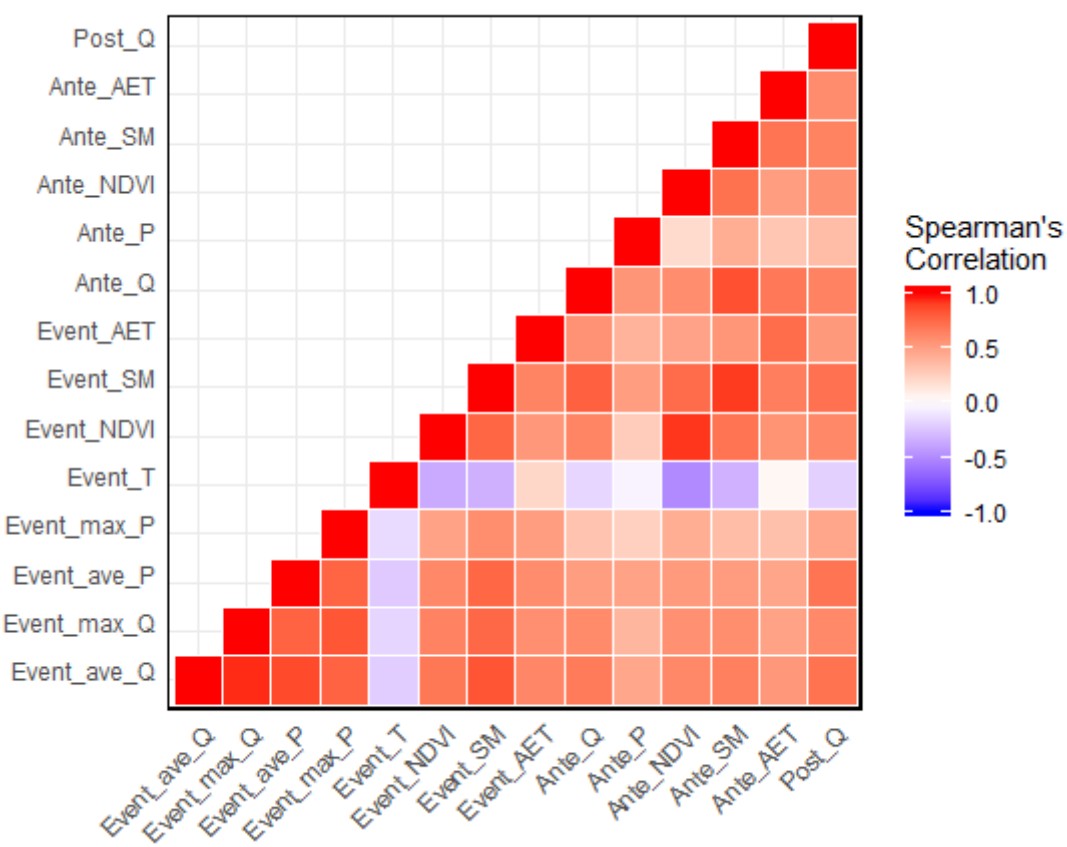

**Figure B1: Spearman's Rank correlation between 14 candidate covariates.**







**Figure B2: Comparison of BMA model coefficient and cumulative model probability (top 100 models) between two clusters for: (a) PN, (b) NH4 and (c) DON. Left - cluster one sites and Right – cluster two sites. The order of predictors on the y-axis was ranked based on the posterior inclusion probability. Each column in the heatmap represents the one specific model (ranked from highest model probability) and the width of the column is normalised by the posterior model probability. The colour indicates the direction of the coefficients, red – negative and blue – positive. Note: the coefficient value was averaged across the posterior median value of site-specific coefficient within each cluster (effect size, $\theta_{n,j}$, in Eq. (6)).**







**Figure B3: Comparison of BMA model coefficient and cumulative model probability (top 100 models) between two clusters for: (a) DOP, (b) PP and (c) EC. Left - cluster one sites and Right – cluster two sites. The order of predictors on the y-axis was ranked based on the posterior inclusion probability. Each column in the heatmap represents the one specific model (ranked from highest model probability) and the width of the column is normalised by the posterior model probability. The colour indicates the direction of the coefficients, red – negative and blue – positive. Note: the coefficient value was averaged across the posterior median value of site-specific coefficient within each cluster (effect size, $\theta_{n,j}$, in Eq. (6)).**





**Figure B4: Distribution of median of site-level coefficients for all plausible models in BMA. (a) PN, (b) NH₄ and (c) DON. Only predictors with PIP > 0.8 are included. For each specific model structure, the coefficient value of a predictor was the median of site-specific coefficient across all sites (effect size, $\theta_{n,j}$, in Eq. (6). The distribution of this value thus represents the probability of the model (PMP), as well as variability in the same predictor across different sites. Note: black dots indicate the median; grey**





vertical lines indicate 95% CI and blue coloured vertical lines indicates 50% CI. The definition of abbreviation of each predictor can be found in Table 3.


Figure B5: Distribution of median of site-level coefficients for all plausible models in BMA. (a) DOP, (b) PP and (c) EC. Only predictors with PIP > 0.8 are included. For each specific model structure, the coefficient value of a predictor was the median of





site-specific coefficient across all sites (effect size, $\theta_{n,j}$, in Equation 6). The distribution of this value thus represents the probability of the model (PMP), as well as variability in the same predictor across different sites. Note: black dots indicate the median; grey vertical lines indicate 95% CI and blue coloured vertical lines indicates 50% CI. The definition of abbreviation of each predictor can be found in Table 3.




**Figure B6: The comparisons of distribution of posterior inclusion probability of individual predictors derived from 1,000 subsampled BMA runs. The interpretation of boxplot is the same as Figure 9.** *Note*: **colour represents different clusters: blue - Cluster and red - Cluster two. The definition of abbreviation of each predictor can be found in Table 3.**






**Figure B7: The comparisons of distribution of posterior inclusion probability of individual predictors derived from 1,000 subsampling BMA. The interpretation of boxplot is the same as Figure 3.** *Note*: **colour represents different clusters: blue - Cluster and red - Cluster two.**


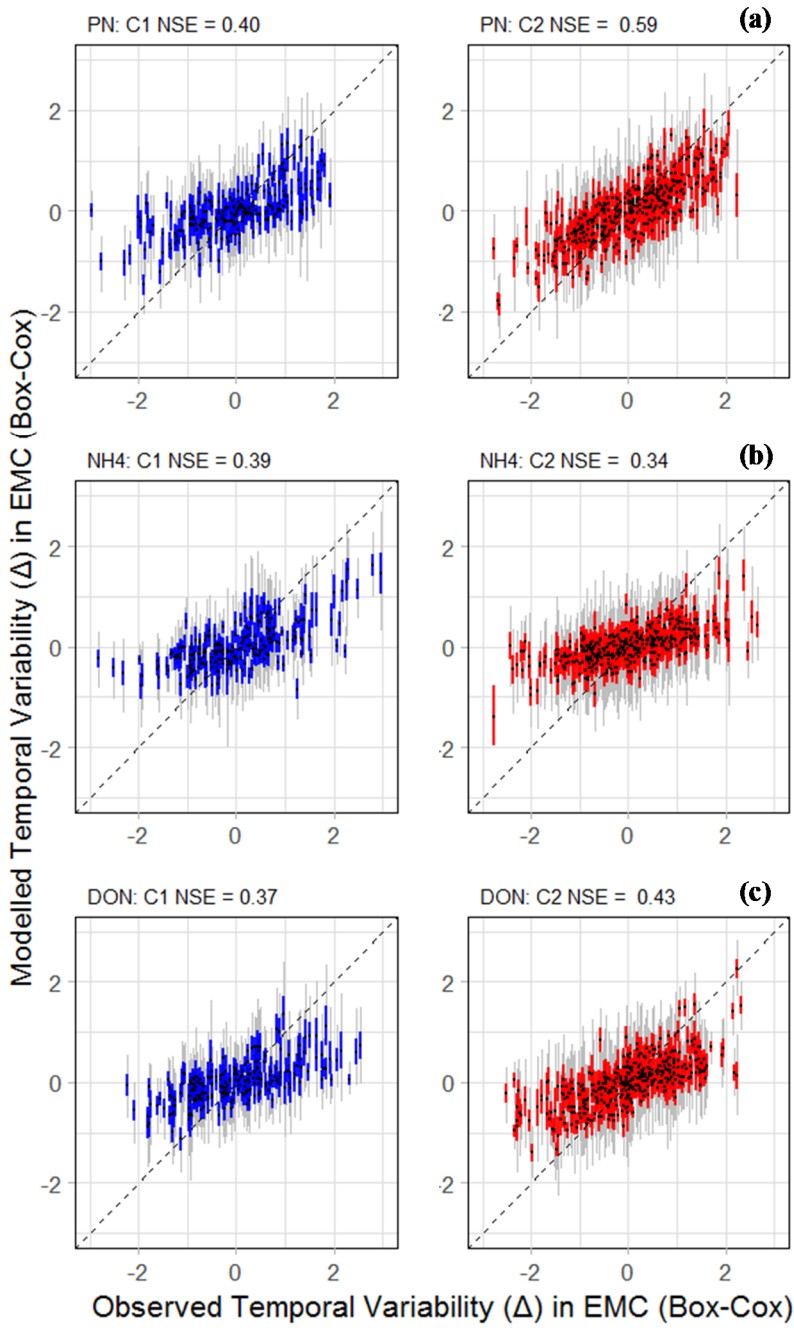

**Figure B8: Performance of the BMA models of the temporal variability of nine constituents across 32 sites, represented by prediction intervals from BMA and observed Box-Cox EMC across two clusters of sites for: (a) PN; (b) NH$_4$ and (c) DON. The NSE values are calculated based on predictions within group- (cluster) level.** *Note*: **black dots are the prediction median; grey vertical lines are the 95% CI and coloured vertical lines indicates 50% CI: blue - Cluster and red - Cluster two. The dashed black lines is the 1:1 relationship.**



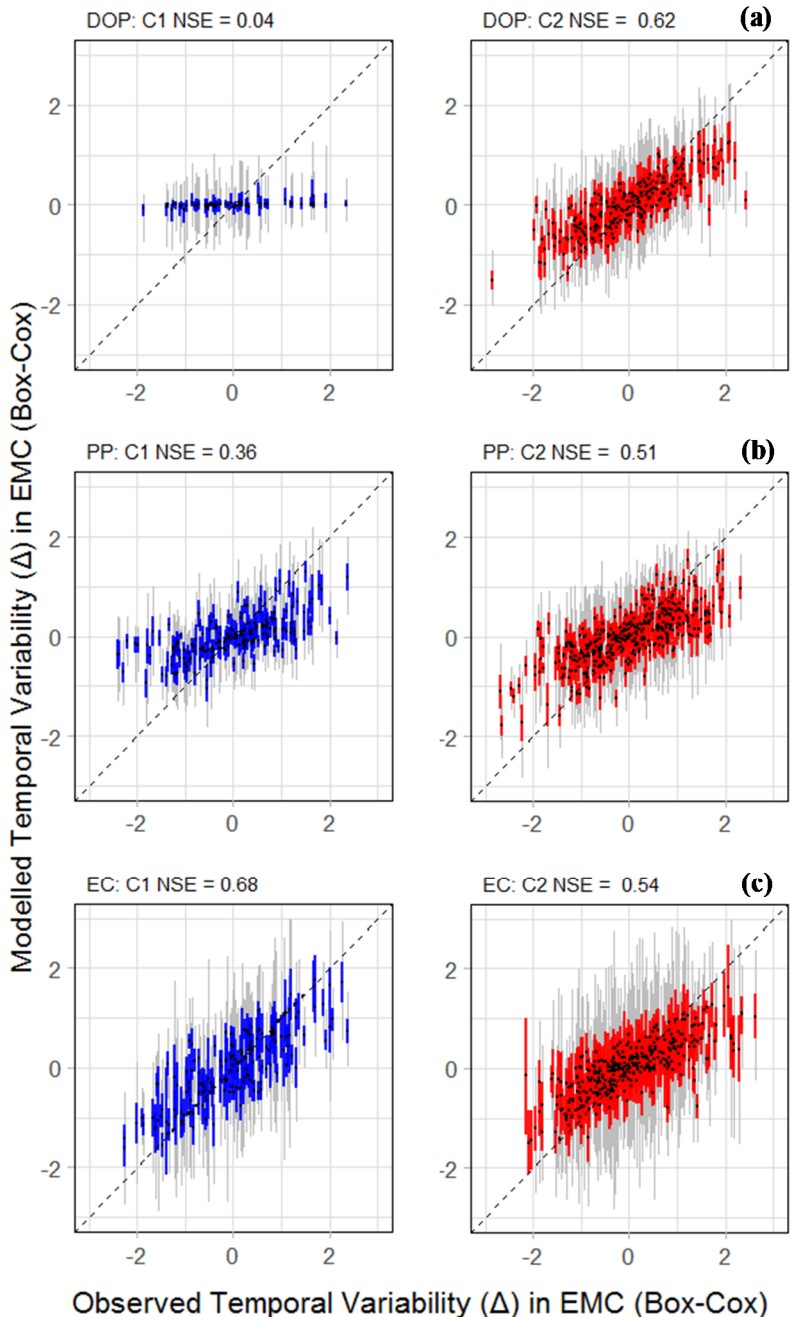

**Figure B9: Performance of the BMA models of the temporal variability of nine constituents across 32 sites, represented by prediction intervals from BMA and observed Box-Cox EMC across two clusters of sites for: (a) DOP; (b) PP and (c) EC. The NSE**

**values are calculated based on predictions within group- (cluster) level. *Note*: black dots are the prediction median; grey vertical lines are the 95% CI and coloured vertical lines indicates 50% CI: blue - Cluster and red - Cluster two. The dashed black lines is the 1:1 relationship.**



**Figure B10: Histograms showing distribution of residuals of nine constituents from BMA predictions. Red – Cluster one; Blue – Cluster two.**





**Figure B11: Relationship between residual in median of BMA prediction of TSS and 14 candidate covariates in BMA. Note, difference colours indicate two clusters: Red – Cluster one; Blue – Cluster two.**







**Figure B12: Relationship between residual in median of BMA prediction of PN and 14 candidate covariates in BMA. Note, difference colours indicate two clusters: Red – Cluster one; Blue – Cluster two.**








**Figure B13: Relationship between residual in median of BMA prediction of NO$_X$ and 14 candidate covariates in BMA. Note, difference colours indicate two clusters: Red – Cluster one; Blue – Cluster two.**








**Figure B14: Relationship between residual in median of BMA prediction of NH₄ and 14 candidate covariates in BMA. Note, difference colours indicate two clusters: Red – Cluster one; Blue – Cluster two.**






**Figure B15: Relationship between residual in median of BMA prediction of DON and 14 candidate covariates in BMA. Note, difference colours indicate two clusters: Red – Cluster one; Blue – Cluster two.**






**Figure B16: Relationship between residual in median of BMA prediction of FRP and 14 candidate covariates in BMA. Note, difference colours indicate two clusters: Red – Cluster one; Blue – Cluster two.**







**Figure B17: Relationship between residual in median of BMA prediction of DOP and 14 candidate covariates in BMA. Note, difference colours indicate two clusters: Red – Cluster one; Blue – Cluster two.**







**Figure B18: Relationship between residual in median of BMA prediction of PP and 14 candidate covariates in BMA. Note, difference colours indicate two clusters: Red – Cluster one; Blue – Cluster two.**








**Figure B19: Relationship between residual in median of BMA prediction of EC and 14 candidate covariates in BMA. Note, difference colours indicate two clusters: Red – Cluster one; Blue – Cluster two.**




## Appendix C - Table

**Table C1. Description of 32 sites in the GBR catchments**

| NRM | Site ID | River and site name | Latitude/° | Longitude/° | Catchment area / km² |
|---|---|---|---|---|---|
| Cape York | 105107A | Normanby River at Kalpowar Crossing | -14.9185 | 144.2100 | 12934 |
| Wet tropics | 110001D | Barron River at Myola | -16.7998 | 145.6121 | 1945 |
| Wet tropics | 110002A | Barron River at Mareeba | -17.0022 | 145.4293 | 836 |
| Wet tropics | 110003A | Barron River at Picnic Crossing | -17.2591 | 145.5386 | 228 |
| Wet tropics | 1110056 | Mulgrave River at Deeral | -17.2075 | 145.9264 | 785 |
| Wet tropics | 1111019 | Russell River at East Russell | -17.2672 | 145.9544 | 524 |
| Wet tropics | 1120049 | North Johnstone River at Old Bruce Hwy Bridge (Goondi) | -17.5059 | 145.9920 | 959 |
| Wet tropics | 112004A | North Johnstone River at Tung Oil | -17.5456 | 145.9325 | 925 |
| Wet tropics | 112101B | South Johnstone River at Upstream Central Mill | -17.6106 | 145.9789 | 400 |
| Wet tropics | 113006A | Tully River at Euramo | -17.9936 | 145.9411 | 1450 |
| Wet tropics | 113015A | Tully River at Tully Gorge National Park | -17.7727 | 145.6507 | 482 |
| Wet tropics | 116001F | Herbert River at Ingham | -18.6328 | 146.1427 | 8581 |
| Burdekin | 119101A | Barratta Creek at Northcote | -19.6923 | 147.1688 | 753 |
| Burdekin | 120001A | Burdekin River at Home Hill | -19.6436 | 147.3958 | 129939 |
| Burdekin | 120002C | Burdekin River at Sellheim | -20.0078 | 146.4369 | 36290 |
| Burdekin | 120301B | Belyando River at Gregory Development Rd. | -21.5423 | 146.8656 | 35410 |
| Burdekin | 120302B | Cape River at Taemas | -20.9996 | 146.4271 | 16070 |
| Burdekin | 120310A | Suttor River at Bowen Developmental Road | -21.5375 | 147.0424 | 10760 |
| Mackay Whitsunday | 124001B | O'Connell River at Stafford's Crossing | -20.6526 | 148.5730 | 342 |
| Mackay Whitsunday | 1240062 | O'Connell River at Caravan Park | -20.5664 | 148.6117 | 825 |
| Mackay Whitsunday | 125013A | Pioneer River at Dumbleton Pump Station | -21.1441 | 149.0753 | 1485 |
| Mackay Whitsunday | 126001A | Sandy Creek at Homebush | -21.2831 | 149.0228 | 326 |
| Fitzroy | 1300000 | Fitzroy River at Rockhampton | -23.3175 | 150.4819 | 139159 |
| Fitzroy | 130206A | Theresa Creek at Gregory Highway | -23.4292 | 148.1514 | 8485 |
| Fitzroy | 130302A | Dawson River at Taroom | -25.6376 | 149.7901 | 15850 |
| Fitzroy | 130504B | Comet River at Comet Weir | -23.6125 | 148.5514 | 16460 |
| Burnett Mary | 136002D | Burnett River at Mt Lawless | -25.5447 | 151.6549 | 29360 |
| Burnett Mary | 136004A | Jones Weir HW | -25.5948 | 151.2964 | 21700 |
| Burnett Mary | 136014A | Burnett River at Ben Anderson Barrage Head Water | -24.8896 | 152.2922 | 32891 |
| Burnett Mary | 136094A | Burnett River at Jones Weir (TW) | -25.5948 | 151.2974 | 21700 |





| Burnett Mary | 136106A | Burnett River at Eidsvold | -25.4023 | 151.1033 | 7117 |
|---|---|---|---|---|---|
| Burnett Mary | 138014A | Mary River at Home Park | -25.7683 | 152.5274 | 6845 |

**Table C2. Number of EMCs for each constituent**

| Cluster | TSS | PN | NO$_X$ | NH$_4$ | DON | FRP | DOP | PP | EC |
|---|---|---|---|---|---|---|---|---|---|
| One | 225 | 207 | 218 | 217 | 215 | 210 | 66 | 186 | 174 |
| Two | 381 | 370 | 372 | 370 | 373 | 372 | 231 | 366 | 354 |
| % of event monitored | 43 | 41 | 42 | 42 | 42 | 41 | 21 | 39 | 37 |


**Table C3. Posterior inclusion probability of individual predictor derived from BMA on two clusters of sites.**

| Predictor | TSS | | PN | | NO$_X$ | | NH$_4$ | | DON | | FRP | | DOP | | PP | | EC | |
|---|---|---|---|---|---|---|---|---|---|---|---|---|---|---|---|---|---|---|
| | Cluster one | Cluster two | Cluster one | Cluster two | Cluster one | Cluster two | Cluster one | Cluster two | Cluster one | Cluster two | Cluster one | Cluster two | Cluster one | Cluster two | Cluster one | Cluster two | Cluster one | Cluster two |
| Event_ave_Q | *0.93* | 0.47 | *0.87* | 0.49 | *0.97* | 0.56 | 0.21 | 0.68 | 0.47 | 0.47 | 0.33 | 0.38 | 0.46 | 0.58 | *0.92* | *0.85* | *1.00* | *0.86* |
| Event_max_Q | 0.73 | *1.00* | 0.76 | *1.00* | 0.33 | 0.58 | 0.14 | 0.79 | 0.79 | 0.60 | *0.94* | 0.43 | 0.46 | *0.85* | 0.66 | *0.99* | *1.00* | 0.63 |
| Event_ave_P | *0.92* | *0.92* | *0.98* | *0.92* | 0.32 | 0.07 | 0.25 | 0.51 | 0.52 | 0.16 | 0.79 | *0.96* | 0.45 | 0.24 | *0.86* | *0.82* | 0.67 | *0.90* |
| Event_max_P | 0.24 | 0.48 | 0.24 | 0.29 | 0.47 | 0.01 | 0.10 | 0.13 | 0.31 | 0.17 | 0.68 | 0.44 | 0.41 | 0.27 | 0.67 | 0.15 | *1.00* | *0.96* |
| Event_T | 0.07 | 0.27 | 0.50 | 0.21 | 0.19 | *0.98* | 0.16 | 0.58 | *0.88* | 0.26 | *0.86* | *0.90* | 0.48 | 0.78 | 0.53 | 0.61 | *1.00* | 0.64 |
| Event_NDVI | 0.03 | 0.27 | 0.09 | *0.89* | 0.77 | 0.55 | 0.68 | 0.62 | 0.35 | 0.34 | 0.38 | 0.49 | 0.46 | 0.97 | 0.39 | 0.52 | 0.75 | 0.79 |
| Event_SM | 0.54 | 0.21 | *0.83* | 0.58 | *0.99* | 0.38 | *0.96* | 0.21 | 0.64 | 0.19 | 0.59 | 0.48 | 0.47 | 0.66 | 0.40 | 0.35 | 0.33 | 0.60 |
| Event_AET | 0.13 | 0.07 | 0.12 | *0.90* | 0.61 | *1.00* | 0.68 | 0.57 | 0.43 | *0.86* | 0.57 | 0.38 | 0.51 | 0.87 | 0.17 | *0.81* | 0.33 | 0.10 |
| Ante_Q | 0.23 | 0.18 | 0.76 | 0.76 | 0.15 | *0.98* | 0.30 | 0.37 | 0.56 | 0.25 | 0.36 | *0.86* | 0.47 | 0.17 | 0.25 | 0.12 | 0.33 | 0.59 |
| Ante_P | 0.20 | 0.05 | 0.22 | 0.06 | 0.16 | 0.03 | 0.25 | 0.70 | 0.22 | 0.75 | 0.25 | *0.88* | 0.44 | 0.98 | 0.13 | 0.06 | *0.81* | *0.91* |
| Ante_NDVI | 0.23 | *1.00* | 0.56 | *1.00* | *0.86* | *1.00* | *0.99* | *0.89* | 0.47 | *1.00* | *0.97* | *0.93* | 0.67 | *1.00* | 0.44 | *1.00* | 0.33 | 0.61 |
| Ante_SM | 0.13 | 0.74 | 0.38 | *0.90* | 0.79 | *1.00* | 0.63 | *0.96* | *0.83* | *0.99* | *0.90* | *0.95* | 0.50 | *0.89* | 0.19 | 0.59 | *1.00* | 0.70 |
| Ante_AET | 0.09 | *0.81* | 0.27 | 0.31 | 0.31 | 0.60 | 0.20 | 0.72 | 0.33 | 0.10 | 0.42 | 0.48 | 0.42 | 0.30 | 0.14 | 0.61 | 0.33 | *1.00* |
| Post_Q | 0.41 | 0.07 | 0.21 | 0.27 | 0.18 | *1.00* | 0.16 | 0.77 | 0.66 | *0.80* | 0.17 | *0.81* | 0.42 | 0.10 | 0.32 | 0.37 | *1.00* | 0.63 |

*Note:* Posterior inclusion probability >= 0.8 in italic.