# Peer review of "A Bayesian approach to understanding the key factors influencing temporal variability in stream water quality: a case study in the Great Barrier Reef catchments"

_Hydrology and Earth System Sciences, 2020_

## Author Comment (AC1)

**Responses to Comments on "A Bayesian approach to understanding the key factors influencing temporal variability in stream water quality: a case study in the Great Barrier Reef catchments" (Referee #1)**

**Anonymous Referee #1 Received and published on 10 Feb 2021.**

**Our responses are in blue and proposed manuscript revisions underlined.**

**General comment**

This manuscript presents a Bayesian modeling approach to understanding factors affecting temporal variability in stream water quality. Overall, I think the manuscript is well written and will become a worthwhile contribution to the hydrological community after moderate revisions. Below I provide some comments to the author, which I hope can help improve the manuscript.

The authors acknowledge the referee's positive comment and the recognition of contribution of this study. The constructive comments will help us improve our manuscript after revision. We provide detailed responses to your comments and our proposed manuscript revisions in the subsequent sections.

**Specific comment**

1. The authors have made it explicit that the current work follows previous study investigating water quality variability in the same region (Liu et al., 2018). There are also other publications from these authors, e.g., Guo et al., 2019, 2020. The discussion section seems not provide much comparison or synthesis of the results from these different but related studies, which appears to be a missed opportunity. I am aware some of these studies focused on temporal patterns and some on temporal patterns. It can potentially become a nice addition to the manuscript and a contribution to the community if the authors can provide some reflection on what different modeling techniques they have used and what new insights on water-quality patterns they have learned from those techniques.

The authors agree with the reviewer and we will incorporate this suggestion. The innovations that this study brings compared to previous studies: 1) Queensland is more event dominated, thus we used event-based water quality data, compared to our previous studies which used monthly water quality data in Victoria; and 2) different modelling methods are used in this study - we used a model averaging approach, rather than a universal modelling framework, which is a more robust approach to understanding the key factors, since the effect of key factors are derived from multiple models.

To address this comment, we will provide more synthesis of the results from this paper, comparing with previous water quality modelling studies. This includes:

1) we will highlight the position of the current study to the broader water quality modelling community in *Introduction*. The focus of this part is to demonstrate different modelling approaches (e.g. simple regression models and process-based hydrological models) (Bartley et al., 2012; Hirsch et al., 2010; Khan et al., 2020; McCloskey et al., 2021), and how these approaches address the spatial and temporal patterns of water quality (Barrientos et al., 2018; Hrachowitz et al., 2016; Kaman et al., 2016; Varanka et al., 2015).

2) we will provide more discussion that specifically compare this study to our previous papers (Guo et al., 2020; Guo et al., 2019) in *Sect.4.2 (Predicting temporal variations in water quality)*.

2. The authors have analyzed nine water quality constituents. While I do appreciate the amount of efforts the authors invested in data analysis and modeling, I wonder if it helps everyone stay focused if the authors were more selective on the constituents. Since a key message from this work is on the different drives of particulate and dissolved constituents, it may be sufficient to select two constituents from each category, as opposed to showing data and results for all nine constituents.

The authors appreciate this suggestion. While it might help the reader more focused when only selected constituents were included in this paper, we would like to keep all nine constituents in the revised manuscript. The reasons that support this decision include:

1) There is a large number of constituents that have been monitored in the GBR water quality monitoring program, but we have reduced the number of constituents for those that have similar patterns. For example, we only retained TSS among TSS, total nitrogen (TN) and total phosphorus (TP).

2) Our analyses are on 9 constituents that are of great concern to the coral reef ecosystem (McCloskey et al., 2017), and could provide a useful comprehensive picture on the overall water quality status and its key temporal drivers. We only consider the 'real parameter' that can be directed measured (expect NOx). This helps to understand full sediment and nutrient budgets exported to the GBR lagoon.

To resolve this comment, in the revised manuscript, the following proposed changes will be made:

1) We will highlight the reason why we select these nine constituents in *Introduction*.

2) We have already been selective on presenting results, focusing on TSS, NOx and FRP i.e. one constituent per category. We will explain in the paper that the results have been simplified and explain our constituent selection rationale in *Sect. 3.1 (Key drivers of temporal variability in water quality)*. In addition, a number of the graphs are already only done for three constituents (e.g., Figures 5 to 8), and we will simplify other figures and tables (e.g., Figure 9 and Table 4) to reflect our focus.

3. Of the two clusters of sites (Figure 2), Cluster 1 sites are quite concentrated, whereas Cluster 2 sites are much more scattered. Also, there seems to be more sites in cluster 2 than cluster 1. I noted that the Bayesian modeling framework was applied to the two clusters independently, I wonder if any of these two aspects (geographical proximity and number of sites) could potentially affect your models and comparison of results between the two clusters. In addition, have you considered developing a single Bayesian model on all sites with the cluster assignment has an explanatory variable?

Thank you for this comment. The clustering results are based on our previous multivariate analysis on the spatial pattern of water quality in the same study area (Liu et al., 2018). We found that distinctive features of the two clusters and their geographic/hydroclimatic differences are responsible for the separation of two clusters of sites. For instance, small wet areas (Cluster 1) near the coast where topography (orography) plays an important role in rainfall generation.  Such geographic features also lead to more dispersed sites in the drier area (Cluster 2).

Furthermore, there are good conceptual reasons for keeping the clusters separate. Based on Liu et al. (2018), results from clustering analyses on spatial patterns of water quality and catchment characteristics were highly correlated, and the two clusters had quite different key explanatory variables. If we put all the sites into the same analysis and just included cluster identity as a random intercept (or even random slope for each explanatory variable) it would skew the choice of explanatory variables for both clusters away from the set achieved in the analysis as it stands. We would end up with same key factors identified for two different clusters, which provides limited information on specific management focuses on two contrasting sets of catchments.

To address this comment, in the revised manuscript, we propose to:

1) provide more details on our previous study on clustering of these catchments in **Sect. 2.1 (Study area)**, and that differences in geographic/hydroclimatic are key factors that distinguished the two clusters of sites. Thus, there are strong practical merits in handling the clusters separately based on the clear contrast between them.

2) improve our description in **Sect. 2.3 (Modelling: driver identification and water quality prediction using multi-model inference)**, to further clarify the reasons behind applying Bayesian model averaging on two different cluster separately.

4. Line 35: In addition to sources, mobilization, and delivery, "transformation" should be included.

Thank you for this comment. We will incorporate this suggestion in the revised manuscript.

5. Section 2.2.3: The authors have quantified the correlation between explanatory variables (Figure B1). Have you considered excluding some variables based on the correlations? If any two variables are highly correlated, it may be wise to keep just one of them in the models.

We have examined the correlation among all explanatory variables, and there are several pairs of variables that are highly correlated (e.g., pre-event NDVI and event NDVI with Spearman's ρ = 0.97). However, it does not necessarily mean they will have similar posterior inclusion probability from BMA (e.g., 1.00 and 0.34 for pre-event NDVI and event NDVI, respectively, for DON in Cluster 2). The BMA can handle the collinearity with shrinking the posterior distribution of the correlated variable to near zero (Posch et al., 2020). This shrinkage effect leads to lower posterior probability of the more complex model (i.e., the model that includes correlated variables), because each extra parameter dilutes the prior density on the pre-existing parameters. Thus, models that include more predictors will have a lower prior probability. Models with additional predictors will be favored only to the extent that their benefit in higher likelihood outweighs their cost in lower prior; however, including correlated variables does not increase the model predictive capacity (Daoud, 2017; Hinne et al., 2020; Kruschke, 2014).

Furthermore, Freckleton (2011) highlighted that when applying model averaging approach, it is not safe to simply exclude correlated variables without due consideration of their likely independent effects. In our case, the high correlation among predictors mainly comes from time lag effects between predictors (e.g., pre-event, event and post-event). The relative importance of these predictors provides strong management indication for future water quality management strategies. Therefore, we would like to keep them all in this analysis.

To resolve this comment, we propose that:

1) highlight that some of the variables are proxies for the same process, and therefore they are closely related in *Sect. 2.2.3 (Explanatory variables)*. We will pay attention to the collinearity issue in the analysis of the results.

2) add more clarification in *Sect. 3.1 (Key drivers of temporal variability in water quality),* that strong correlation between predictors does not necessary mean that the posterior inclusion probability of these factors is similar. In addition, we will provide more discussion on how BMA address the collinearity issue in in *Sect. 3.1.*

6. BMA model coefficients plots (Figure 5 and other related figures in the SM): I found it difficult to compare the patterns across clusters or among constituents because the variables are not displayed in the same order in these panels.

Thank you for this comment. We will incorporate this suggestion in the revised manuscript (e.g., reorder the predictors of Figure 5 and other plots in SM to make sure they follow the same order).

7. Predictive model performance (Section 3.2 and Table 4): The NSE values are not high, some are very low. This seems to limit the utility of the proposed Bayesian approach, which the authors should discuss and defend against.

We agree with the referee that the NSE values are not high, but based on the recommended performance measures from Moriasi et al. (2015), most of the model performance is satisfactory (Table 1), especially for the Cluster 2 models. Generally, low NSE is acceptable for modelling nutrients and sediment compared to hydrology. It is also worth noting that most of water quality models evaluated in Moriasi et al. (2015) are physically-based models (e.g. SWAT, HSPF, WARME), and focusing on individual catchments. However, we used a statistical modelling approached to predict multiple catchments and to identify key factors simultaneously. We agree that the model performance for DOP in Cluster 1 is very poor, and we have provided detailed discussion of this in *Lines 428 to 438* in *Sect. 4.2 (Predicting temporal variations in water quality)*. Therefore, we did not rely on any results for DOP in Cluster 1 when analyzing the results.

*Table 1 Performance statistics for nine constituents for the modelled and observed temporal variability, according to Moriasi et al. (2015).*

| Constituent | Cluster one | Cluster two |
|---|---|---|
| TSS | Indicative | Satisfactory |
| PN | Satisfactory | Satisfactory |
| $NO_X$ | Satisfactory | Good |
| $NH_4$ | Satisfactory | Indicative |
| DON | Satisfactory | Satisfactory |
| FRP | Satisfactory | Satisfactory |
| DOP | Indicative | Good |
| PP | Satisfactory | Satisfactory |
| EC | Satisfactory | Satisfactory |

To address this comment, we propose to:

1) provide additional assessment of model performance based on the recommended performance measures from Moriasi et al. (2015) in *Sect. 4.2 (Predicting temporal variations in water quality).* We will demonstrate that our predictive ability is comparable to other water quality models.

2) provide more discussion in *Sect. 4.2* that we are not inferring any conclusions from the modelling results for DOP in Cluster 1, due to the poor performance.

8. Line 415: Again, the effect is not only on transportation but also on transformation. Specifically, temperature is expected to affect the intensity of biological processes, e.g., denitrification.

Thank you for this comment. We will incorporate this suggestion in the revised manuscript.

**Reference**

Barrientos, Guillermo, & Iroumé, Andrés. (2018). The effects of topography and forest management on water storage in catchments in south-central Chile. *Hydrological Processes, 32*(21), 3225-3240.

Bartley, Rebecca, Speirs, William J, Ellis, Tim W, & Waters, David K. (2012). A review of sediment and nutrient concentration data from Australia for use in catchment water quality models. *Marine pollution bulletin, 65*(4-9), 101-116.

Daoud, Jamal I. (2017). *Multicollinearity and regression analysis.* Paper presented at the Journal of Physics: Conference Series.

Freckleton, Robert P. (2011). Dealing with collinearity in behavioural and ecological data: model averaging and the problems of measurement error. *Behavioral Ecology and Sociobiology, 65*(1), 91-101.

Guo, Danlu, Lintern, Anna, Webb, J Angus, Ryu, Dongryeol, Bende-Michl, Ulrike, Liu, Shuci, & Western, Andrew William. (2020). A data-based predictive model for spatiotemporal variability in stream water quality. *Hydrology and Earth System Sciences, 24*(2), 827-847.

Guo, Danlu, Lintern, Anna, Webb, J Angus, Ryu, Dongryeol, Liu, Shuci, Bende-Michl, Ulrike, Leahy, Paul, Wilson, Paul, & Western, AW. (2019). Key Factors Affecting Temporal Variability in Stream Water Quality. *Water Resources Research, 55*(1), 112-129.

Hinne, Max, Gronau, Quentin F., van den Bergh, Don, & Wagenmakers, Eric-Jan. (2020). A Conceptual Introduction to Bayesian Model Averaging. *Advances in Methods and Practices in Psychological Science, 3*(2), 200-215. doi: 10.1177/2515245919898657

Hirsch, Robert M, Moyer, Douglas L, & Archfield, Stacey A. (2010). Weighted regressions on time, discharge, and season (WRTDS), with an application to Chesapeake Bay river inputs 1. *JAWRA Journal of the American Water Resources Association, 46*(5), 857-880.

Hrachowitz, Markus, Benettin, Paolo, Van Breukelen, Boris M, Fovet, Ophelie, Howden, Nicholas JK, Ruiz, Laurent, Van Der Velde, Ype, & Wade, Andrew J. (2016). Transit times—the link between hydrology and water quality at the catchment scale. *Wiley Interdisciplinary Reviews: Water, 3*(5), 629-657.

Kaman, Harun, Cetin, Mahmut, & Kırda, Cevat. (2016). SPATIAL AND TEMPORAL EVALUATION OF DEPTH AND SALINITY OF THE GROUNDWATER IN A LARGE IRRIGATED AREA IN SOUTHERN TURKEY. *Pakistan Journal of Agricultural Sciences, 53*(2).

Khan, Urooj, Cook, Freeman J, Laugesen, Richard, Hasan, Mohammad M, Plastow, Kevin, Amirthanathan, Gnanathikkam E, Bari, Mohammed A, & Tuteja, Narendra K. (2020). Development of catchment water quality models within a realtime status and forecast system for the Great Barrier Reef. *Environmental Modelling & Software, 132*, 104790.

Kruschke, John. (2014). *Doing Bayesian data analysis: A tutorial with R, JAGS, and Stan*: Academic Press.

Liu, S, Ryu, D, Webb, JA, Lintern, A, Waters, D, Guo, Danlu, & Western, AW. (2018). Characterisation of spatial variability in water quality in the Great Barrier Reef catchments using multivariate statistical analysis. *Marine pollution bulletin, 137*, 137-151. doi: https://doi.org/10.1016/j.marpolbul.2018.10.019

McCloskey, G, Waters, D, Baheerathan, R, Darr, S, Dougall, C, Ellis, R, Fentie, B, & Hateley, L. (2017). Modelling reductions of pollutant loads due to improved management practices in the great barrier reef catchments: Updated methodology and results-technical report for reef report card 2015. *Queensland Department of Natural Resources and Mines, Brisbane, Queensland*.

McCloskey, GL, Baheerathan, R, Dougall, C, Ellis, R, Bennett, FR, Waters, D, Darr, S, Fentie, B, Hateley, LR, & Askildsen, M. (2021). Modelled estimates of fine sediment and particulate nutrients delivered from the Great Barrier Reef catchments. *Marine pollution bulletin, 165*, 112163.

Moriasi, Daniel N, Gitau, Margaret W, Pai, Naresh, & Daggupati, Prasad. (2015). Hydrologic and water quality models: Performance measures and evaluation criteria. *Transactions of the ASABE, 58*(6), 1763-1785.

Posch, Konstantin, Arbeiter, Maximilian, & Pilz, Juergen. (2020). A novel Bayesian approach for variable selection in linear regression models. *Computational Statistics & Data Analysis, 144*, 106881.

Varanka, Sanna, Hjort, Jan, & Luoto, Miska. (2015). Geomorphological factors predict water quality in boreal rivers. *Earth Surface Processes and Landforms, 40*(15), 1989-1999.

---

## Author Comment (AC2)

**Responses to Comments on "A Bayesian approach to understanding the key factors influencing temporal variability in stream water quality: a case study in the Great Barrier Reef catchments" (Referee #2)**

**Anonymous Referee #2

**Our responses are in blue and proposed manuscript revisions underlined.**

**General comment**

Liu et al. conducted an improved Bayesian approach to evaluate the temporal variability in stream water quality and the related key factors. This study aimed to: i) identify the key influencing factors, and ii) predict the temporal variation, taking advantages of multiple locations and multiple water quality monitoring data. In addition, authors divided the study sites into two clusters and analyzed separately, which might avoid potential uncertainty issues caused by a single model, and improve the scientific and reliability of the modelling results.

This study is an interesting topic and generally well written. It contributes to our knowledge of both the further application of the developed Bayesian model framework and the understanding the temporal water quality variability in the Great Barrier Reef catchments. In general, this piece of work could be considered for publication after some unclear concerns were addressed.

Thank you for your comprehensive review and recognition of the study contribution. The constructive comments will help us improve our manuscript after revision. We provide detailed responses to your comments and our proposed manuscript revisions in the subsequent sections.

**Major comment**

1. Section 2.2.2 The authors gave a detailed process of data extraction and processing. Among them, it was noticed that "The start and end points of a specific event were determined by using a local minimum method that calculates the first derivative of the streamflow record (separated from baseflow)". Basing on your data processing method, when can be identified as the start or end points? I think more details of the key standard or parameter maybe better for the readers to further understand your approach.

Thank you for this comment. We agree with the referee that it is a bit ambiguous as to how the event is delineated (i.e., definition of start and end points of events). Here we used an automated approach developed by Tang et al. (2017), which allows us to extract runoff event on the baseflow-free hydrograph, by specifying a set of parameters (e.g., $\beta$ filter coefficient, $ReTh$ difference between two flows to set the local minima for event extraction). As illustrated in Figure 1, once the local minimum T1 is found, the next local minimum T2 is considered as the

first candidate end point. *ReTh* is used to filter out any false end point, which allows the flows at the start and end of an event can be different. This Matlab toolbox directly returns the start and end points of an event, avoid time-consuming and subjective inconsistent outcomes.

[Figure]

Figure 1. An example of selecting the end point of a runoff event (Tang et al., 2017).

To resolve this comment, in the revised manuscript, we will:

1) clarify the method we used to delineate flow events in *Sect. 2.2.2 (Event mean concentration)* and provide the key specifications (i.e., parameters) for running the Hydrun toolbox in the *Appendix*.

2) provide an example hydrograph with start and end points in the *Appendix* (as shown in the figure below).

[Figure]

Figure 2. Delineation of runoff events and estimation of EMCs, based on the hydrograph for 105107A Normanby River at Kalpowar Crossing in the GBR catchments: (a) baseflow separation from continuous streamflow observations; (b) event identification and development of EMC, and 35 runoff events are identified with red dots representing either the start or end of a runoff event; and (c) A zoom in event #9 in 2008.

2. The authors divided the site locations of the GBR catchments into two clusters (wet and dry), and modelled separately. The advantages of the subsequent result are obvious, i.e., pertinence, reliability and so on. However, whether the strong pertinence will reduce the universality of this approach and limit its universal application? And if it is necessary to add the model and discussion of all sites?

Thank you for this comment. Application of the model based on two clusters does not limit the utility of the model. We aim to identify the key factors affecting temporal variability in water quality for two different clusters of sites, but this method can be used anywhere, e.g., a universal application for all sites. There are strong practical and conceptual reasons that we decide to model the two clusters of sites separately (i.e., we have provided detailed

justifications in our reply to the Comment #3 from Referee #1). By doing this, we are not making claims that there are always variables that will be important in such catchments. Therefore, our method is universal, but our results are not.

To address this comment, in the revised manuscript, we propose to:

1) provide more details on our previous study on clustering of these catchments in *Sect. 2.1 (Study area)*, and that differences in geographic/hydroclimatic are key factors that distinguished the two clusters of sites. Thus, there are strong practical merits in handling the clusters separately based on the clear contrast between them.

2) add discussion in *Sect. 4.2 (Predicting temporal variations in water quality)*, to further clarify that applying our modelling approach to two clusters of sites does not limit the utility of the method.

3. The authors targeted nine common water quality indicators, including sediments, nutrients and salinity. But in the nutrients part, they only focused on N and P, without any constituents about Carbon studied. Why? Please explain it.

Water quality parameters included in the Queensland Government's Loads Monitoring Program are those that enter the Great Barrier Reef lagoon from inland catchments, and suspended solids, nutrients and pesticide are the focus of this program. Carbon from inland entering the GBR lagoon is not as harmful as sediments and nutrients for the coral reef ecosystem, thus carbon is not monitored. We could not include carbon in our analysis.

4. 2.2.2 again "The event-mean concentration (EMC) was then calculated for each event that had at least two samples on each of the rising and falling limbs of the hydrograph." Table C2 showed the Number of EMCs for each constituent. So what is the approximate amount of data per event? Why you set two samples as the minimum limitation? whether two samples are too few?

Thanks for your comment. First, we would like to highlight that we set the minimum of two on both rising and falling limbs of the hydrograph, then we have minimum 4 samples per event. Second, in Bartley et al. (2012), they review water quality data in Australian catchments and use minimum 3 samples over an event as threshold to calculate EMC (Tables 4 to 6, Bartley et al. (2012)). Therefore, our four samples per event is above the standard set by that review paper. In addition, we have calculated that, on average, there are 14 samples per event across nine constituents (ranging from 12 for DOP to 16 for EC), therefore our calculated EMCs are reliable.

To address this comment, we propose:

1) clarify the choice of four samples (2 on both rising and falling limbs) per event in ***Sect. 2.2.2 (Event mean concentration).***

2) provide a summary table (Table 1) that indicates number of samples per event for each constituent in the ***Appendix***.

*Table 1. Average number of samples per event for each constituent*

| TSS | PN | NO$_X$ | NH$_4$ | DON | FRP | DOP | PP | EC |
|-----|-----|--------|--------|-----|-----|-----|-----|-----|
| 15 | 14 | 14 | 14 | 14 | 15 | 12 | 14 | 16 |

5. I also noticed that you normalized the data of each event first and then calculate the Event mean concentration. If this process is necessary?

Thank for your comment. We think the referee may misunderstood our method. We normalized the EMC rather than the original water quality data. We will revise our manuscript to clarify this in ***Sect. 2.2.2 Event mean concentration***. Also, normalization of the predictand is necessary to facilitate the fitting process and fulfill the statistical assumption of our model; we use Bayesian linear regression with the response variable sampled from a normal distribution (Atkinson, 2020; Castillo et al., 2015; Hoeting et al., 2002). We will incorporate in ***Sect. 2.2.2 (Event mean concentration)***.

**Minor comments:**

6. Fig 1d  six_NRM regions.

Thank you for this comment. We will incorporate this suggestion in the revised manuscript.

7. Table 1 Delete the comma at the end of the sentence in the item "Land use/land cover" of Cluster 2.

Thank you for this comment. We will incorporate this suggestion in the revised manuscript.

8. L274 Delete the full name of "MCMC", which has appeared in the line 261.

Thank you for this comment. We will incorporate this suggestion in the revised manuscript.

**Reference**

Atkinson, Anthony B. (2020). The box-cox transformation: Review and extensions. *Statistical science*.

Bartley, Rebecca, Speirs, William J, Ellis, Tim W, & Waters, David K. (2012). A review of sediment and nutrient concentration data from Australia for use in catchment water quality models. *Marine pollution bulletin, 65*(4-9), 101-116.

Castillo, Ismaël, Schmidt-Hieber, Johannes, & Van der Vaart, Aad. (2015). Bayesian linear regression with sparse priors. *Annals of Statistics, 43*(5), 1986-2018.

Hoeting, Jennifer A, Raftery, Adrian E, & Madigan, David. (2002). Bayesian variable and transformation selection in linear regression. *Journal of Computational and Graphical Statistics, 11*(3), 485-507.

Tang, Weigang, & Carey, Sean K. (2017). HydRun: A MATLAB toolbox for rainfall–runoff analysis. *Hydrological Processes, 31*(15), 2670-2682.

---

## Author Response (AR1)

**Responses to Comments on "A Bayesian approach to understanding the key factors influencing temporal variability in stream water quality: a case study in the Great Barrier Reef catchments" (Referee #1)**

**Anonymous Referee #1 Received and published on 10 Feb 2021.**

**Our responses are in blue and changes in the revised manuscript underlined.**

**General comment**

This manuscript presents a Bayesian modeling approach to understanding factors affecting temporal variability in stream water quality. Overall, I think the manuscript is well written and will become a worthwhile contribution to the hydrological community after moderate revisions. Below I provide some comments to the author, which I hope can help improve the manuscript.

The authors acknowledge the referee's positive comment and the recognition of contribution of this study. The constructive comments will help us improve our manuscript after revision. We provide detailed responses to your comments and changes in the revised manuscript in the subsequent sections.

**Specific comment**

1. The authors have made it explicit that the current work follows previous study investigating water quality variability in the same region (Liu et al., 2018). There are also other publications from these authors, e.g., Guo et al., 2019, 2020. The discussion section seems not provide much comparison or synthesis of the results from these different but related studies, which appears to be a missed opportunity. I am aware some of these studies focused on temporal patterns and some on temporal patterns. It can potentially become a nice addition to the manuscript and a contribution to the community if the authors can provide some reflection on what different modeling techniques they have used and what new insights on water-quality patterns they have learned from those techniques.

The authors agree with the reviewer and we will incorporate this suggestion. The innovations that this study brings compared to previous studies: 1) Queensland is more event dominated, thus we used event-based water quality data, compared to our previous studies which used monthly water quality data in Victoria; and 2) different modelling methods are used in this study - we used a model averaging approach, rather than a single-model approach, which is a more robust approach to understanding the key factors, since the effect of key factors are derived from multiple models.

To address this comment, we have provided more synthesis of the results from this paper, comparing with previous water quality modelling studies. This includes:

1) we have highlighted the position of the current study to the broader water quality modelling community in ***Introduction***:

*Line 59:* "Process-based and statistical modelling approaches have been widely used to investigate water quality temporal dynamics in response to changes in the abovementioned environmental factors (Fu et al., 2019; Wellen et al., 2015). Process-based water quality models use complex mass-balance structures, describing the water quality source, mobilisation and transport processes (Abbott et al., 1986; Merritt et al., 2003). They are typically based on hydrological and biogeochemical processes that can affect the generation and transport of pollutants into receiving waters. These models (e.g., Soil and Water Assessment Tool – SWAT, and Source Catchments) have been applied to assess the impact of land use management and climate on sediment and pollutant concentrations (Arnold et al., 2005; Francesconi et al., 2016; Qi et al., 2018), optimise water management and delivery for agriculture, industry and environmental uses (Ly et al., 2019), and estimate pollutant generation, loss and transport processes (Jayakrishnan et al., 2005; McCloskey et al., 2021). However, the complexity of process-based models results in intensive data and calibration requirements, and large-scale application has been limited (Abbaspour et al., 2015; Arnold et al., 2005). These models may also have large uncertainties in the interpretability of the parameters and their characterization of the effects of specific processes (Wade et al., 2002), such as denitrification in streams (Filoso et al., 2004)."

*Line 72:* "On the other hand, statistical water quality models have a relatively simple mathematical structure, an ability to quantify predictive uncertainty (Kasiviswanathan et al., 2013; Srivastav et al., 2007) and low requirement for *a priori* information on distinct processes (Afed Ullah et al., 2018; Letcher et al., 2002; Mainali et al., 2019; Schwarz et al., 2006). However, existing statistical water quality modelling studies have limitations."

*Line 85:* "This study attempted to address these knowledge gaps in statistical water quality models, taking advantage of event-based water quality monitoring data from the Great Barrier Reef (GBR) catchments in northern Australia, where land-derived pollutants have posed threats to ecosystem of the GBR lagoon (Brodie et al., 2012; McKergow et al., 2005b; Waterhouse et al., 2017). We address the limitations in statistical water quality models by using: 1) Bayesian hierarchical modelling was used to investigate water quality temporal variation, whichallowed the prediction of water quality in multiple catchments, as well as simultaneously quantifying parameter uncertainty (Gelman et al., 2013; Rode et al., 2010; Webb et al., 2009); and 2) Bayesian model averaging (BMA) approaches were used to identify the relative importance of the different environmental factors and provide multi-model weighted predictions, which have been shown to better quantify the uncertainty arising from model selection (Höge et al., 2019; Raftery et al., 1997; Wang et al., 2012)."

2) we have provided more discussion that specifically compares this study to our previous papers (Guo et al., 2020; Guo et al., 2019) in *Sect.4.2 (Predicting temporal variations in water quality)*:

*Line 483:* "The modelling performance in this study was generally higher than our previous studies (i.e., Guo et al. (2019) and Guo et al. (2020)). This improved performance can be attributed to:

(1) difference in water quality monitoring data

Rivers in Queensland are more event dominated, thus we used event-based water quality data, compared to our previous studies which used monthly water quality data in Victoria. The uncertainty in event-based water quality samples have less impact on modelling performance because we considered the variability in streamflow when developing EMCs in this study (Chen et al., 2017; Lessels et al., 2015; Letcher et al., 2002).

(2) difference in modelling methods

Here, we used a model averaging approach that considered model predictions from multiple candidate models, rather than a single-model approach that was used in our previous studies (Guo et al. 2019, 2020). This approach is a more robust approach to providing predictions because the predictions consider the model selection uncertainty (Höge et al., 2019; Raftery et al., 1997)."

2. The authors have analyzed nine water quality constituents. While I do appreciate the amount of efforts the authors invested in data analysis and modeling, I wonder if it helps everyone stay focused if the authors were more selective on the constituents. Since a key message from this work is on the different drives of particulate and dissolved constituents, it may be sufficient to select two constituents from each category, as opposed to showing data and results for all nine constituents.

The authors appreciate this suggestion. While it might help the reader more focused when only selected constituents were included in this paper, we would like to keep all nine constituents in the revised manuscript. Our analyses are on the nine constituents that are of great concern to the coral reef ecosystem (McCloskey et al., 2017), and could provide a useful comprehensive picture on the overall water quality status and its key temporal drivers. We have also constrained the variables to only the 'real parameter' that can be directed measured (expect for NOx). This helps to understand full sediment and nutrient loads being exported to the GBR lagoon.

To resolve this comment, in the revised manuscript, the following changes have been made:

1) We have highlighted the reason why we select these nine constituents in ***Introduction***.

***Line 93:*** "We targeted nine common water quality indicators, including sediments, nutrients and salinity. This is a subset of the constituents that have been monitored in the GBR water quality monitoring program. Our analyses are conducted on constituents that are of great concern to the coral reef ecosystem (McCloskey et al., 2017), and could provide a useful comprehensive picture on the overall water quality status. Finally, we have constrained the variables to only the 'real parameter' that can be directed measured (with the exception of $NO_X$), which helps to understand full sediment and nutrient loads being exported to the GBR lagoon."

2) We have already been selective on presenting results, focusing on TSS, NOx and FRP i.e. one constituent per category. We have explained in the paper that the results have been simplified and explain our constituent selection rationale in ***Sect. 3.1 Key drivers of temporal variability in water quality***.

*Line 320:* "Results from here on will focus mainly on TSS, NO$_X$ and FRP, due to their impacts on the marine receiving environment. Results for the other six constituents are in Appendix.

In addition, a number of the graphs are already only done for three constituents (e.g., Figs. 5 to 8), and we have moved Fig. 9 and Table 4 to Appendix (as Fig. B10 and Table C5) to reflect our focus.

3. Of the two clusters of sites (Fig. 2), Cluster 1 sites are quite concentrated, whereas Cluster 2 sites are much more scattered. Also, there seems to be more sites in cluster 2 than cluster 1. I noted that the Bayesian modeling framework was applied to the two clusters independently, I wonder if any of these two aspects (geographical proximity and number of sites) could potentially affect your models and comparison of results between the two clusters. In addition, have you considered developing a single Bayesian model on all sites with the cluster assignment has an explanatory variable?

Thank you for this comment. The clustering decisions are based on our previous multivariate analysis on the spatial pattern of water quality in the same study area (Liu et al., 2018). We found that distinctive features of the two clusters and their geographic/hydroclimatic differences are responsible for the separation of two clusters of sites. For instance, small wet areas (Cluster 1) near the coast where topography (orography) plays an important role in rainfall generation.  Such geographic differences also lead to more dispersed sites in the drier area (Cluster 2). However, geographical proximity and number of sites do not affect the model results and comparison of the results between the two clusters. This is because we are not making claims that there are always variables that will be important in such catchments. Such that, our modelling method is universal, but our results are not.

Furthermore, there are good conceptual reasons for keeping the clusters separate. Based on Liu et al. (2018), results from clustering analyses on spatial patterns of water quality and catchment characteristics were highly correlated, and the two clusters had quite different key explanatory variables. If we put all the sites into the same analysis and just included cluster identity as a random intercept (or even random slope for each explanatory variable), it would make it more difficult to identify a universal set of key explanatory variables that represent both clusters. We would end up with same key factors identified for two different clusters, which provides limited information on specific management focuses on two contrasting sets of catchments.

set of key explanatory variables that represent both clusters and likely increase the uncertainty of the coefficients too. The analysis would identify the same key factors identified for the two different clusters. It is important to consider and model these clusters separately so that we can better inform how water quality can be managed in these separate environmental conditions."

4. Line 35: In addition to sources, mobilization, and delivery, "transformation" should be included.

We have incorporated this in the revised manuscript as follows:

*Line 33:* "These spatial and temporal variations are the result of complex interactions between three key pollutant processes in catchments, namely, sources (e.g., atmospheric deposition or anthropogenic inputs), mobilisation (e.g., detachment from the sources), delivery (e.g., transport from sources to receiving waters) and transformation (e.g., biogeochemical processes) (Granger et al., 2010; Harris, 2001; Lintern et al., 2018a)."

5. Section 2.2.3: The authors have quantified the correlation between explanatory variables (Figure B1). Have you considered excluding some variables based on the correlations? If any two variables are highly correlated, it may be wise to keep just one of them in the models.

We have examined the correlation among all explanatory variables, and there are several pairs of predictors that are highly correlated (e.g., pre-event NDVI and event NDVI with Spearman's $\rho = 0.97$). However, it does not necessarily mean they will have similar posterior inclusion probability from BMA (e.g., 1.00 and 0.34 for pre-event NDVI and event NDVI, respectively, for DON in Cluster 2). The BMA can handle the collinearity with shrinking the posterior distribution of the one of the correlated predictors towards zero (Posch et al., 2020). This shrinkage effect leads to a lower posterior probability of a more complex model that includes correlated variables, because each extra predictor dilutes the prior density of the existing predictor that it correlates with. Such more complex model is unlikely to be selected, unless the loss in posterior probability can be outweighed by the gain in achieving a higher likelihood (Daoud, 2017; Hinne et al., 2020; Kruschke, 2014).

Furthermore, Freckleton (2011) highlighted that when applying the model averaging approach, it is not safe to simply exclude correlated variables without due consideration of their likely independent effects. In our case, the high correlation among predictors mainly comes from time lag effects between predictors (e.g., pre-event, event and post-event). The relative importance of these predictors provides strong management indication for future water quality management strategies. Therefore, we have not removed any correlated predictors in this analysis.

To resolve this comment, we have revised our manuscript as follows:

*Line 209:* "Some of the variables are proxies for the same process, and thus some paired predictors are highly correlated (e.g., pre-event NDVI and event NDVI with Spearman's $\rho = 0.97$). Freckleton (2011) highlighted that when applying the model averaging approach, it is not safe to simply exclude correlated variables without due consideration of their likely independent effects. In our case, the high correlation among predictors mainly comes from time lag effects between predictors (e.g., pre-event, event and post-event). The relative importance of these predictors provides strong management indication for future water quality management strategies. Therefore, we have not removed any correlated predictors in this analysis. It is likely that different model structures result in similar predictive performance (discussed in the analysis of the results, i.e., Sect. 3.1)."

*Line 307:* "It is also worth noting that strong correlations between predictors does not necessary mean that the posterior inclusion probability of these factors is similar (e.g., 1.00 and 0.34 for pre-event NDVI and event NDVI, respectively, for DON in Cluster 2). The BMA can handle the collinearity with shrinking the posterior distribution of inclusion probability of one of the correlated variables towards zero (Nakagawa et al., 2011; Posch et al., 2020; Walker, 2019). This shrinkage effect leads to a lower posterior probability of a more complex model that includes correlated variables, because each extra predictor dilutes the prior density of the existing predictor that it correlates with. Such more complex model is unlikely to be selected, unless the loss in posterior probability can be outweighed by the gain in achieving a higher likelihood (Daoud, 2017; Hinne et al., 2020; Kruschke, 2014)."

6. BMA model coefficients plots (Figure 5 and other related figures in the SM): I found it difficult to compare the patterns across clusters or among constituents because the variables are not displayed in the same order in these panels.

Thank you for this comment. We have incorporated this in the revised manuscript (e.g., reordered the predictors of Figs 5, B3 and B4 to make sure they follow the same order). We have also removed the description of 'The order of predictors on the y-axis was ranked based on the posterior inclusion probability' from the captions of these figures. Please see Fig. 5 below as an example.

[Figure]

*Figure 5: Comparison of BMA model coefficients and cumulative model probabilities (only the first 100 models ranked according to the highest probability are shown) between Cluster 1 ("wet" - left) and Cluster 2 ("dry" - right) sites for [a] TSS, [b] NO$_X$ and [c] FRP. Each column in the heatmap represents the one specific model (ranked from highest model probability from left to right) and the width of the column is normalised by the posterior model probability (i.e., the widest columns indicate models with the largest increase in probability compared to the next most probable model). The colour indicates the direction of the coefficients: red = negative; blue = positive. The coefficient value was averaged across the posterior median value of the site-specific coefficient within each cluster (effect size, $\theta_{n,j}$, in Equation 6); the definition of the abbreviations of each predictor on the y-axis are in Table 3.*

7. Predictive model performance (Section 3.2 and Table 4): The NSE values are not high, some are very low. This seems to limit the utility of the proposed Bayesian approach, which the authors should discuss and defend against.

We agree with the referee that the NSE values are not high, but based on the recommended performance measures from Moriasi et al. (2015), model performance is satisfactory (Table 1), especially for the Cluster 2 models. We agree that the model performance for DOP in Cluster 1 is very poor, and we have provided detailed discussion of this in *Lines 471 to 481* in *Sect. 4.2 Predicting temporal variations in water quality*. Therefore, we did not rely on any results for DOP in Cluster 1 when analyzing the results.

*Table C7. Performance statistics for nine constituents for the modelled and observed temporal variability, according to Moriasi et al. (2015).*

| Constituent | Cluster one | Cluster two |
| --- | --- | --- |
| TSS | Indicative | Satisfactory |
| PN | Satisfactory | Satisfactory |
| NO$_X$ | Satisfactory | Good |
| NH$_4$ | Satisfactory | Indicative |
| DON | Satisfactory | Satisfactory |
| FRP | Satisfactory | Satisfactory |
| DOP | Indicative | Good |
| PP | Satisfactory | Satisfactory |
| EC | Satisfactory | Satisfactory |

To address this comment, we have provide additional assessment of model performance based on the recommended performance measures from Moriasi et al. (2015) in *Sect. 3.2 Predictive performance* as follows:

*Line 369:* "According to model performance criteria recommended by Moriasi et al. (2015), model performance is satisfactory (Table C7), especially for the Cluster 2 models. Generally, low NSE is acceptable for modelling nutrients and sediment compared to hydrology. It is also worth noting that, in contrast to the models developed here, most of the water quality models evaluated in Moriasi et al. (2015) are process-based models and focusing on individual catchments."

In addition, we have provided more discussion in *Sect. 4.2* on the results of DOP in Cluster 1 as follows:

*Line 481:* "Therefore, we did not infer any conclusions from the modelling results of DOP in Cluster 1 due to the poor modelling performance."

8. Line 415: Again, the effect is not only on transportation but also on transformation. Specifically, temperature is expected to affect the intensity of biological processes, e.g., denitrification.

Thank you for this comment. We have incorporated this suggestion in the revised manuscript as follows:

*Line 456:* "Temperature is one controlling factor that affects pollution transformation (Barnard et al., 2005). For instance, temperature has a direct impact on the activity of microorganisms, which affects the intensity of biological processes such as denitrification (Wakelin et al., 2011)."

**Anonymous Referee #2

**Our responses are in blue and changes in the revised manuscript underlined.**

**General comment**

Liu et al. conducted an improved Bayesian approach to evaluate the temporal variability in stream water quality and the related key factors. This study aimed to: i) identify the key influencing factors, and ii) predict the temporal variation, taking advantages of multiple locations and multiple water quality monitoring data. In addition, authors divided the study sites into two clusters and analyzed separately, which might avoid potential uncertainty issues caused by a single model, and improve the scientific and reliability of the modelling results.

This study is an interesting topic and generally well written. It contributes to our knowledge of both the further application of the developed Bayesian model framework and the understanding the temporal water quality variability in the Great Barrier Reef catchments. In general, this piece of work could be considered for publication after some unclear concerns were addressed.

Thank you for your comprehensive review and recognition of the study contribution. The constructive comments will help us improve our manuscript after revision. We provide detailed responses to your comments and changes in the revised manuscript in the subsequent sections.

**Major comment**

1. Section 2.2.2   The authors gave a detailed process of data extraction and processing. Among them, it was noticed that "The start and end points of a specific event were determined by using a local minimum method that calculates the first derivative of the streamflow record (separated from baseflow)". Basing on your data processing method, when can be identified as the start or end points? I think more details of the key standard or parameter maybe better for the readers to further understand your approach.

Thank you for this comment. To clarify, for event delineation, we have used an automated approach developed by Tang et al. (2017) as a Matlab toolbox *HydRun*, which allowed us to extract runoff event on the baseflow-free hydrograph. This approach required a set of parameters (e.g., *β* filter coefficient, *ReTh* difference between two flows to set the local minima for event extraction). The delineation process is as illustrated in Fig. R1: once the local minimum T1 is found, the next local minimum T2 is considered as the first candidate end point. *ReTh* is used to filter out any false end point, which allows the flows at the start and end of an event can be different. This Matlab toolbox directly returns the start and end points of an event, avoid time-consuming and subjective inconsistent outcomes. These parameters are determined based on recommended values from literature (Garzon-Garcia et al., 2016;

Ladson et al., 2013; Zhang et al., 2017), as well as manual review of all event hydrographs to ensure overall consistency.

[Figure]

*Figure R1. An example of selecting the end point of a runoff event (Tang et al., 2017).*

To resolve this comment, in the revised manuscript, we have clarified the method we used to delineate flow events in ***Sect. 2.2.2 (Event mean concentration)***, provided the key specifications (i.e., parameters) for running the Hydrun toolbox, and an example hydrograph with start and end points in the ***Appendix***:

***Line 148:*** "An automated hydrograph analysis tool – HydRun (Tang et al., 2017) was used to delineate runoff events. This approach allowed us to extract runoff event on the baseflow-free hydrograph, by specifying a set of parameters (e.g., β filter coefficient, *ReTh* difference between two flows to set the local minima for event extraction). This toolbox directly returned the start and end points of an event, thereby avoiding time-consuming and subjective inconsistent outcomes. The key parameters used for HydRun Toolbox are provided in Table C2 (Appendix C) and an example hydrograph output is provided in Fig. B1. These parameters are determined based on recommended values from literature (Garzon-Garcia et al., 2016; Ladson et al., 2013; Zhang et al., 2017), as well as manual review of all event hydrographs to ensure overall consistency."

*Table C2. Parameters for running the Hydrun toolbox*

| Filter coefficient | Pass for baseflow separation | Peak discharge threshold | Return ratio | Smooth coefficient | Minimum duration |
|---|---|---|---|---|---|
| 0.975 | 3 | 100 | 0.01 | 6 | 24 |

[Figure]

*Figure B1. Delineation of runoff events and estimation of EMCs, based on the hydrograph for 105107A Normanby River at Kalpowar Crossing in the GBR catchments: (a) baseflow separation from continuous streamflow observations; (b) event identification and development of EMC, and 35 runoff events are identified with red dots representing either the start or end of a runoff event; and (c) A zoom in event #9 in 2008.*

2. The authors divided the site locations of the GBR catchments into two clusters (wet and dry), and modelled separately. The advantages of the subsequent result are obvious, i.e., pertinence, reliability and so on. However, whether the strong pertinence will reduce the universality of this approach and limit its universal application? And if it is necessary to add the model and discussion of all sites?

Thank you for this comment. Application of the model based on two clusters does not limit the utility of the model. There are strong practical and conceptual reasons that we decide to model the two clusters of sites separately (i.e., we have provided detailed justifications in our reply to the **Comment #3** from **Referee #1**). We aimed to identify the key factors affecting temporal variability in water quality for two different clusters of sites, but this method can be used anywhere, e.g., a single modelling framework for all sites.  By doing this, we are not making claims that there are always variables that will be important in such catchments. Therefore, our method is universal, but our results are not.

To address this comment, in the revised manuscript, we have provided more details on our previous study on clustering of these catchments in **Sect. 2 Materials and methods**:

*Line 118:* "We found that differences in geographic/hydroclimatic catchment characteristics (Fig. 2 [b], [c] and [d]) are the key factors that distinguished the two clusters of sites (e.g., small wet areas (Cluster 1) near the coast where topography (orography) plays an important role in rainfall generation) (Liu et al., 2018). Such geographic differences also lead to more dispersed sites in the drier area (Cluster 2).

*Line 219:* "There are strong practical merits in handling the clusters separately. Previous results from clustering analyses on spatial patterns of water quality and catchment characteristics were highly correlated, and that the two clusters had quite different key explanatory variables (Liu et al., 2018). If all the sites were pooled into the same analysis, it would make it more difficult to identify a universal set of key explanatory variables that represent both clusters and likely increase the uncertainty of the coefficients too. The analysis would identify the same key factors identified for the two different clusters. It is important to consider and model these clusters separately so that we can better inform how water quality can be managed in these separate environmental conditions."

In addition, we have added more discussion in **Sect. 4.2 Predicting temporal variations in water quality**, to further clarify that applying our modelling approach to two clusters of sites does not limit the utility of the method:

*Line 500:* "In addition, as discussed in Sect. 2.3, due to strong practical and conceptual reasons, our modelling framework was applied to two clusters of sites separately. However, this method can be used anywhere, e.g., a single modelling framework for all sites. Thus, we are not making claims that there are always variables that will be important in such catchments. Our method is universal, but our results are not."

3. The authors targeted nine common water quality indicators, including sediments, nutrients and salinity. But in the nutrients part, they only focused on N and P, without any constituents about Carbon studied. Why? Please explain it.

Water quality parameters included in the Queensland Government's Loads Monitoring Program (**Sect. 2.1.1**) are those that enter the Great Barrier Reef lagoon from inland catchments, and suspended solids, nutrients and pesticide are the focus of this program. Carbon from inland entering the GBR lagoon is not as harmful as sediments and nutrients for the coral reef ecosystem, thus carbon is not monitored. Whilst including carbon in our analyses would have been valuable, we could not include carbon in our analysis due to the lack of data.

4. 2.2.2 again "The event-mean concentration (EMC) was then calculated for each event that had at least two samples on each of the rising and falling limbs of the hydrograph." Table C2 showed the Number of EMCs for each constituent. So what is the approximate amount of data per event? Why you set two samples as the minimum limitation? whether two samples are too few?

Thanks for your comment. First, we would like to highlight that we set the minimum of two on both rising and falling limbs of the hydrograph, then we have minimum 4 samples per event. Second, in Bartley et al. (2012), they review water quality data in Australian catchments and a minimum of 3 samples over an event as threshold to calculate EMC (Tables 4 to 6, Bartley et al. (2012)). Therefore, our four samples per event is above the standard set by that review paper. In addition, we have calculated that, on average, there are 14 samples per event across the nine constituents (ranging from 12 for DOP to 16 for EC), therefore our calculated EMCs are reliable.

To address this comment, we have clarified the choice of four samples (2 on both rising and falling limbs) per event in **Sect. 2.2.2 Event mean concentration**, and provided a summary table (Table C3) that indicates number of samples per event for each constituent in the **Appendix**:

**Line 156:** "Thus, for each EMC, a minimum of 4 samples was achieved, which is above the standard (3 samples per event) set by Bartley et al. (2012). On average, there were 14 samples per event across the nine constituents (ranging from 12 for DOP to 16 for EC, Table C3). This ensured that the water quality dynamics over a runoff event were reasonably well-captured, and that the derived EMCs were reliable."

Table C3. Average number of samples per event for each constituent

| TSS | PN | NO$_X$ | NH$_4$ | DON | FRP | DOP | PP | EC |
|-----|-----|--------|--------|-----|-----|-----|-----|-----|
| 15 | 14 | 14 | 14 | 14 | 15 | 12 | 14 | 16 |

5. I also noticed that you normalized the data of each event first and then calculate the Event mean concentration. If this process is necessary?

Thank for your comment. We think the referee may have misunderstood our method. We normalized the EMC rather than the original water quality data. Normalization of the predictand (EMC) is necessary to facilitate the fitting process and fulfill the statistical assumption of our model; we use Bayesian linear regression with the response variable sampled from a normal distribution (Atkinson, 2020; Castillo et al., 2015; Hoeting et al., 2002).

To address this comment, we have revised our manuscript to clarify this in **Sect. 2.2.2 Event mean concentration** as follows:

**Line 169:** "The derived EMCs (i.e., rather than the individual water quality samples) were Box-Cox transformed to improve the symmetry of the response variable (Box et al., 1964). The normalization of the predictand is necessary to facilitate the fitting process and fulfil the statistical assumption of our model. This is because we use a Bayesian linear regression with the response variable sampled from a normal distribution (Sect. 2.3.1.) (Atkinson, 2020; Castillo et al., 2015; Hoeting et al., 2002)."

*Minor comments:*

6. Fig 1d  six_NRM regions.

Thank you for this comment. We believe this comment is related to Fig. 2 rather than Fig. 1. We have revised the figure as below (updated Fig. 2 [d]).

[Figure]

*Figure 2: Spatial information of the GBR catchments in northeast of Australia: [a] site locations showing two groups based on clustering analysis of spatial variability in time-averaged water quality (Liu et al., 2018); [b] topographic elevation (250 m resolution) (Geoscience Australia, 2008); [c] annual average rainfall (Bureau of Meteorology, 2012), and [d] updated Köppen-Geiger climate zone classification (Peel et al., 2007).*

7. Table 1 Delete the comma at the end of the sentence in the item "Land use/land cover" of Cluster 2.

Thank you for this comment. We have incorporated this in the revised manuscript (updated Table 1).

8. L274 Delete the full name of "MCMC", which has appeared in the line 261.

Thank you for this comment. We have incorporated this in the revised manuscript (*Line 285*).

**Reference**

Atkinson, Anthony B. (2020). The box-cox transformation: Review and extensions. *Statistical science*.

Bartley, Rebecca, Speirs, William J, Ellis, Tim W, & Waters, David K. (2012). A review of sediment and nutrient concentration data from Australia for use in catchment water quality models. *Marine pollution bulletin, 65*(4-9), 101-116.

Castillo, Ismaël, Schmidt-Hieber, Johannes, & Van der Vaart, Aad. (2015). Bayesian linear regression with sparse priors. *Annals of Statistics, 43*(5), 1986-2018.

Daoud, Jamal I. (2017). *Multicollinearity and regression analysis.* Paper presented at the Journal of Physics: Conference Series.

Freckleton, Robert P. (2011). Dealing with collinearity in behavioural and ecological data: model averaging and the problems of measurement error. *Behavioral Ecology and Sociobiology, 65*(1), 91-101.

Garzon-Garcia, A, Wallace, Rohan, Huggins, Rae, Turner, Ryan DR, Smith, Rachael, Orr, David, Ferguson, Ben, Gardiner, Richard, Thomson, Belinda, & Warne, Michael. (2016). Total suspended solids, nutrient and pesticide loads (2013–2014) for rivers that discharge to the Great Barrier Reef.

Guo, Danlu, Lintern, Anna, Webb, J Angus, Ryu, Dongryeol, Bende-Michl, Ulrike, Liu, Shuci, & Western, Andrew William. (2020). A data-based predictive model for spatiotemporal variability in stream water quality. *Hydrology and Earth System Sciences, 24*(2), 827-847.

Guo, Danlu, Lintern, Anna, Webb, J Angus, Ryu, Dongryeol, Liu, Shuci, Bende‐Michl, Ulrike, Leahy, Paul, Wilson, Paul, & Western, AW. (2019). Key Factors Affecting Temporal Variability in Stream Water Quality. *Water Resources Research, 55*(1), 112-129.

Hinne, Max, Gronau, Quentin F., van den Bergh, Don, & Wagenmakers, Eric-Jan. (2020). A Conceptual Introduction to Bayesian Model Averaging. *Advances in Methods and Practices in Psychological Science, 3*(2), 200-215. doi: 10.1177/2515245919898657

Hoeting, Jennifer A, Raftery, Adrian E, & Madigan, David. (2002). Bayesian variable and transformation selection in linear regression. *Journal of Computational and Graphical Statistics, 11*(3), 485-507.

Kruschke, John. (2014). *Doing Bayesian data analysis: A tutorial with R, JAGS, and Stan*: Academic Press.

Ladson, Anthony Richard, Brown, R, Neal, B, & Nathan, R. (2013). A standard approach to baseflow separation using the Lyne and Hollick filter. *Australasian Journal of Water Resources, 17*(1), 25-34.

Liu, S, Ryu, D, Webb, JA, Lintern, A, Waters, D, Guo, Danlu, & Western, AW. (2018). Characterisation of spatial variability in water quality in the Great Barrier Reef catchments using multivariate statistical analysis. *Marine pollution bulletin, 137*, 137-151. doi: https://doi.org/10.1016/j.marpolbul.2018.10.019

McCloskey, G, Waters, D, Baheerathan, R, Darr, S, Dougall, C, Ellis, R, Fentie, B, & Hateley, L. (2017). Modelling reductions of pollutant loads due to improved management practices in the great barrier reef

catchments: Updated methodology and results-technical report for reef report card 2015. *Queensland Department of Natural Resources and Mines, Brisbane, Queensland*.

Moriasi, Daniel N, Gitau, Margaret W, Pai, Naresh, & Daggupati, Prasad. (2015). Hydrologic and water quality models: Performance measures and evaluation criteria. *Transactions of the ASABE, 58*(6), 1763-1785.

Posch, Konstantin, Arbeiter, Maximilian, & Pilz, Juergen. (2020). A novel Bayesian approach for variable selection in linear regression models. *Computational Statistics & Data Analysis, 144*, 106881.

Tang, Weigang, & Carey, Sean K. (2017). HydRun: A MATLAB toolbox for rainfall–runoff analysis. *Hydrological Processes, 31*(15), 2670-2682.

Zhang, Junlong, Zhang, Yongqiang, Song, Jinxi, & Cheng, Lei. (2017). Evaluating relative merits of four baseflow separation methods in Eastern Australia. *Journal of Hydrology, 549*, 252-263.

---

## Author Response (AR2)

Responses to Comments on "A Bayesian approach to understanding the key factors influencing temporal variability in stream water quality: a case study in the Great Barrier Reef catchments" (Editor)

**General comment**

Comments to the Author:

Thank you for the revised manuscript and for clearly documenting your responses to the reviewers.

I am pleased to recommend the article for publication.

The manuscript would benefit from some additional minor editing, mostly relating to the way responses to reviewers have been incorporated into the original text. For example, on L34 you currently write, "interactions between three key pollutant processes..." and then list four processes. Please take advantage of the copy-editing stage to carefully re-read and correct the proofs prior to publication.

The authors acknowledge the editor's effort and time on our manuscript, and the positive comment from the editor is much appreciated. We have revised the section in Line34 as follows:

L34"These spatial and temporal variations are the result of complex interactions between four key pollutant processes in catchments, namely,….".

We have also removed the materials in Appendix to Supplement, which is more suitable for these materials to be presented.